# Pedestrians' safety using projected time-to-collision to electric scooters

**Alireza Jafari** [1] **& Yen-Chen Liu** [1] ✉

Safety concern among electric scooter riders drives them onto sidewalks, endangering pedestrians and making them uncomfortable. Regulators' solutions are inconsistent and conflicting worldwide. Widely accepted pedestrian safety metrics may lead to converging solutions. Adapting the time-to-collision from car traffic safety, we define projected time-to-collision and experimentally study pedestrians' objective and subjective safety. We design isolated and crowd experiments using e-scooter-to-pedestrian interactions to assess the impact of various factors on objective safety. In addition, we conducted a pedestrian survey to relate the subjective safety and the metric. We report a strong correlation between subjective safety and the projected time-to-collision when agents face each other and no relation when the e-scooter overtakes a pedestrian. As a near-miss metric correlated with pedestrian comfort, projected time-to-collision is implementable in policy-making, urban architecture, and e-scooter design to enhance pedestrian safety.

As an emerging micro-mobility transportation, electric scooters are attracting public attention rapidly. Liu et al.[1] categorize the growing interest into recreational or joy riding and non-recreational uses. Non-recreational applications mainly address the first/last mile of transport and connect transportation hubs to final destinations such as schools and parks (see ref. 2). In a systematic review, Zhang et al.[3] expressed concerns about the sudden spike in popularity and the challenges our cities face regarding pedestrian safety and infrastructure efficiency.

Feeling vulnerable, these new road users are reluctant to move side by side with the cars in the streets (see ref. 4). Thus, they join sidewalks endangering pedestrians, the primary sidewalk users, and causing undue discomfort, as reported by Che et al.[5] and Šucha et al.[6]. Cicchino et al.[7] analysed a set of documented injuries and reported that 58% of e-scooter injuries happened on sidewalks compared to 23% on roads; the severity of the accidents was significantly higher on roads. Therefore, the policies addressing the issue must act with tact. On the one hand, a total ban on the e-scooters' entrance to the sidewalks drives them toward roads and puts the riders at serious risk, and on the other hand, letting them freely roam on sidewalks endangers pedestrian safety and comfort.

The regulators' seemingly arbitrary and sometimes contradictory solutions show that more investigation is required to handle the

dilemma safely. For example, some cities like Madrid or Santa Monica entirely ban e-scooter entrance onto the sidewalks, whereas others like Paris or Detroit have no regulations (see ref. 8). The former may lead to severe or life-threatening rider injuries, while the latter may lead to more accidents besides pedestrians' discomfort. In addition, Asensio et al.[9] report that restrictions on micro-mobility vehicles cause a significant increase in travel time and eventually lead to congestion and emission concerns. Simulations like the studies of Coretti Sanchez et al.[10,11] can evaluate solutions' impact on users. But first, the studies must quantify the users' safety to form a basis for comparing various solutions.

Regarding objective safety, regulations, and solutions backed by data tend to be more effective and acceptable to all stakeholders. Thus, field experiments must look for the origin of the safety concerns in causes, including vehicle design, riders' behaviour, or incompatible infrastructure (see ref. 12). To study if the root cause is with the vehicle or the human, Dozza et al.[13], experimentally compared the acceleration and deceleration of e-scooters and bicycles and concluded that e-scooters braking is not as effective as bicycles. However, they are more manoeuvrable (due to a lower centre of mass) and comfortable, suggesting collision avoidance by steering away may be a better strategy than just braking. Our experiments study riders' collision avoidance using combined braking-steering

[1]Department of Mechanical Engineering, National Cheng Kung University, No. 1, Dasyue Rd, East District, Tainan, Taiwan. ✉e-mail: yliu@mail.ncku.edu.tw

and evaluate pedestrian safety by comparing a time-to-collision variant with response time.

In addition, the study of pedestrians' subjective safety regarding e-scooters also provides insights into comfortable integration. Liu et al.[14] used accumulated acceleration and social force as pedestrian safety and comfort metrics during Monte Carlo simulations. Moreover, Kuo et al.[15] studied pedestrians' comfort levels when facing personal mobility devices. They concluded that the comfort drops noticeably when the personal mobility device moves faster than 4 m/s with a lateral distance shorter than 50–70 cm. James et al.[16] used a survey to evaluate pedestrians' risk perception during interactions with e-scooter riders and e-cyclists. The pedestrians were less comfortable with e-scooters, potentially because they were less familiar with the personal mobility device.

Like e-scooters, mobile robots on sidewalks also affect pedestrian safety (see ref. [17]). Thus, data-driven frameworks extendable to other agent types are favourable. For example, Zhou et al.[18] suggested that if the mobile robot's movement patterns are human-like, the neighbouring pedestrians perceive it as natural and, thus, feel more comfortable. Shiomi et al.[19] employed a social force model to imitate human collision avoidance patterns in a shopping mall. As a result, the human experience around the mobile robot improved without real-time knowledge of people's discomfort. To quantify people's discomfort, Hasegawa et al. [20] experimentally studied pedestrians' danger perception toward personal mobility vehicles (PMV) using a pedestrian and a Segway-type PMV. They recorded the pedestrian's subjective danger using questionnaires and compared it with a social-force-model-based estimation of discomfort.

Psychological discomfort arises from the agent's appearance and movement (see ref. [21]). As the primary movement states, this research uses relative position and velocity, combines them in Time-to-Collision (TTC), and studies its variation through field experiments as an objective and subjective safety metric. Originally, TTC appeared in transportation safety research to determine near-miss car traffic events. Hayward[22] used trained but unaware human observers to watch video recordings of dangerous events and then collected their opinions on the danger levels. The minimum TTC successfully predicted most of the perceived danger levels by the observers. In addition, Sun and Frost[23] analysed pigeons' brain response to looming objects and highlighted the role of TTC in neurons' firing patterns.

Years later, traffic safety research still recognizes TTC as a critical measure of an accident's imminence[24]. Archer[25] named the minimum TTC as a conflicts' primary time-based severity surrogate measure. Later, case-specialized definitions of TTC emerged. Zhang et al. [26] addressed less studied pedestrian-involved collisions with cars focusing on a variant of TTC. They defined the time difference to collision and evaluated its performance in classifying dangerous situations using recorded video data of actual zebra crossing. In addition, Zhang et al.[27], developed an algorithm for collision risk assessment of a car facing multiple pedestrians. They calculated a potential collision area using vehicles' and pedestrians' linear and angular velocities. The method selects the pedestrian with minimum TTC to the area as the most vulnerable target. Moreover, Nie et al.[28] reported four pedestrian actions to avoid collisions with oncoming cars, i.e., stepping back, rushing forward, diagonal walking and no reaction. The study emphasizes the required time for pedestrians to avoid collision in the perception–decision–action mechanism. We believe these results point to the role of a direction-dependent TTC variant in walkers' collision avoidance mechanisms. In addition, Schwarz[29] used the time of closest approach (TCA), suggesting computation algorithms using bounding boxes. We compare our proposed metric to TCA.

Overall, the safety metrics for car-to-car accidents are well-developed. However, when pedestrians are involved, the safety metrics are often overlooked and less studied. Among the few studies on pedestrians' safety, studies focusing on walkers' subjective safety or

comfort are even rarer, perhaps because it is more difficult to quantify. In the context of e-scooter interactions with pedestrians, objective safety refers to the likelihood of physical harm or injury resulting from the interaction, whereas subjective safety refers to the perceived feeling of safety or comfort by the pedestrian. Objective safety can be measured through descriptive statistics of TTC or its variants, critical distance, and velocity, while subjective safety can be assessed through surveys or questionnaires that measure perceived safety or comfort.

The novelty of this work lies in the experimental study of projected time-to-collision (PTTC), i.e., a TTC variant and an objective safety metric, with various factors including the sidewalk width, the agent type, and the interaction format, i.e., facing or overtaking. In addition, we report a strong correlation between the reported discomfort or the subjective safety and the PTTC providing functions to estimate neighbouring pedestrians' feelings around e-scooters. Moreover, we evaluate the results with an e-scooter moving through multiple pedestrians. The presence of other pedestrians shows the extent of the single encounter results' generalizability. We believe that PTTC contributes to both objective and subjective safety, and controlling it benefits the integration of new modes of transportation into public spaces.

## Results

We compare PTTC or $T_p$, with the required response time and observe that in most cases, there is a time span during the e-scooter-pedestrian interaction in which the PTTC becomes smaller than the response time; for the definitions (see the "Methods" section). If another contributing factor, for example, hazardous road features, pedestrians in blind spots, or a brief distraction, happens during this period, there is no time to prevent the potential accident.

Dozza et al.[13] experimentally measured the response time, i.e., the time from a stop command to the beginning of the e-scooter deceleration, including the rider and e-scooter internal braking delays. The median of the reported results is 0.75 s with lower and upper quartiles of 0.55 and 0.95 s. Since PTTC may still decrease after the braking starts, $T_p$ is usually smaller than the response time. Nevertheless, in the graphical representations in this paper, for overtaking cases when the interaction is unilateral, we highlight the areas with $T_p < 0.55$ s, $0.55$ s $< T_p < 0.95$ s, and $0.95$ s $< T_p$ with red, yellow, and green showing danger, alarm and safe zones, respectively. Moreover, unlike the response time, $T_p$ results from cooperation between the two interacting agents and is determined by both reactions. Thus, when the interactions are bilateral (facing cases), we reduce the limits to 0.4 and 0.7 s because the cooperation of the agents lowers the chance of an accident. Note that although the highlights are rooted in experimental results, they only give the reader a better understanding of the boxplots and are not verified limits.

### Observations

The experiment procedure and summarized data are in the "Methods" section. We employ notched box plots to compare our observations of different interaction settings when changing a specific parameter. Notched boxplots are a graphical representation used to compare the medians of two or more groups. The notches in the boxplot represent the 95% confidence intervals of the medians. If the notches of the two groups do not overlap, it indicates that the medians are significantly different at the 5% significance level ($p$-value $< 0.05$) based on a two-sample $t$-test. Thus, they help the reader identify differences in medians between groups and assess statistical significance. For a detailed comparison, see the Supplementary Tables.

We quantify how near a miss is using $T_p$ and study how it changes with sidewalk width, agent type, and interaction format. Using the data provided by interaction settings A and C, Fig. 1a displays the $T_p$ for widths ranging from 1.3 to 3.0 m. The plot reveals that the interaction between an e-scooter and a pedestrian falls within the danger zone

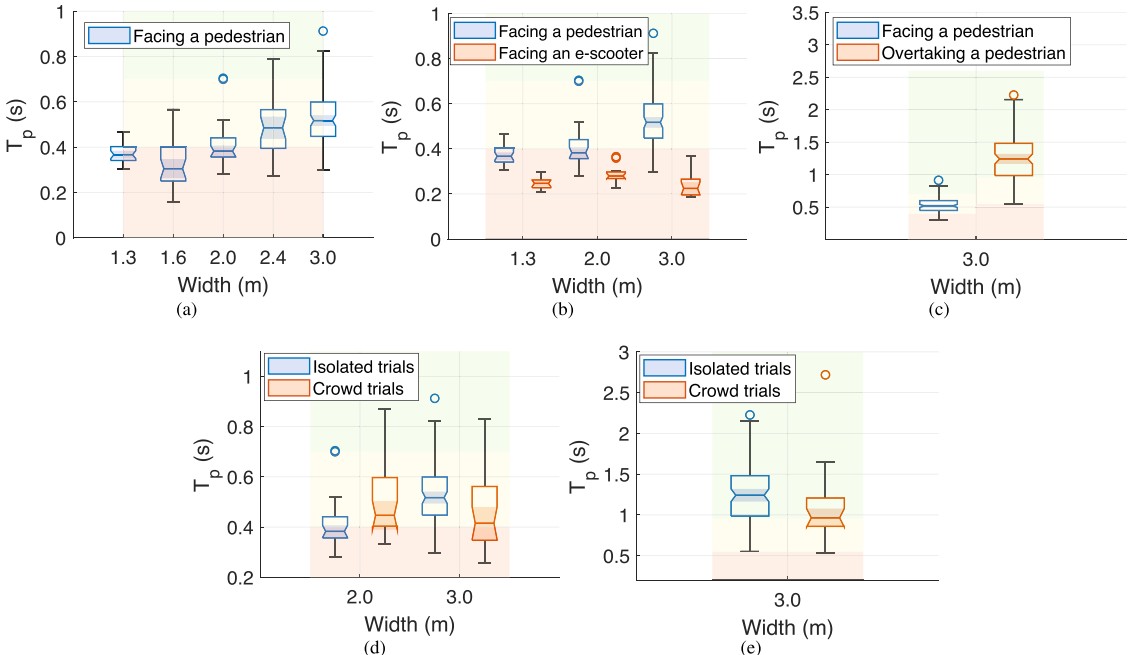

**Fig. 1 | Analysis of $T_p$ correlation with sidewalk width, agent type, and interaction format.** The collected $T_p$ from Table 3, **a** versus width when the e-scooter faces a pedestrian using interaction settings A and C; **b** versus width and agent type when the e-scooter faces an e-scooter or a pedestrian using interaction settings A, C, and D (due to the limited number of participants, **b** serves as a general guideline); **c** versus interaction format when the e-scooter faces or overtakes a pedestrian on a 3.0 m wide sidewalk using interaction settings A and B. The coloured areas stand for danger (red), alarm (yellow), and safe zones (green). The collected $T_p$ from isolated trials next to the collected $T_p$ from crowd trials when **d** an e-scooter faces a pedestrian, interaction settings A and E; **e** an e-scooter overtakes a pedestrian, interaction settings B and E. Interaction setting E's pedestrians are divided into two facing and overtaking categories and are separately displayed in the corresponding figure. The scale changes across the sub-figures. For box plot details, see the Supplementary Table 1. For a detailed statistical comparison, see the Supplementary Tables 2–7.

when the sidewalk width is smaller than 2.0 m. The most severe case occurs at 1.6 m rather than 1.3 m. An explanation is that the $T_p$ is higher at 1.3 m because the riders significantly reduce velocity on such a narrow sidewalk. The sidewalk is wider when the width is 1.6 m or even 2.0 m, allowing the riders to move faster. However, in doing so, they may overestimate the available extra space and ride so quickly that it creates an even more dangerous interaction than the narrower case. When the sidewalk width is wider than 2.0 m, the rider has enough space to move comfortably, ensuring a safer interaction. Moreover, the statistically insignificant difference between widths of 2.4 and 3.0 m hints that, in one-on-one interactions, broader sidewalks are not necessarily safer or more comfortable.

Figure 1b contains $T_p$ calculated when an e-scooter faces a pedestrian or another e-scooter on sidewalks with widths of 1.3, 2.0, and 3.0 m. Interaction settings A, C, and D provide the agent's type plot data. Because the number of trials in Interaction setting D is limited, Fig. 1b serves as a general guideline. Due to the higher relative velocity, when the e-scooters face each other, the interaction is well inside the danger zone considering their response time, even though the collision avoidance is cooperative. Moreover, when the width is 3.0 m, the $T_p$ is lower than narrower sidewalks. The mentioned explanation regarding the overestimation of the extra space happens for e-scooter–e-scooter interaction, too, but only on a broader sidewalk. When facing pedestrians, e-scooter riders ride carelessly when the width is 1.6 m, whereas when facing other e-scooters, they do it when the width is 3.0 m.

Figure 1c compares the $T_p$ for the interaction format study, i.e., the e-scooter faces or overtakes a pedestrian on a 3.0 m wide sidewalk using the collected data through interaction settings A or B. The danger zones are different due to the difference in interaction cooperativeness. Whereas the facing case is cooperative, the overtaking case is non-cooperative since the pedestrian can not see the e-scooter

and thus can not contribute to collision avoidance. Therefore, the restrictions on safe $T_p$ are assumed to be tighter when the e-scooter approaches the pedestrian from behind. The $T_p$ is significantly different for the cases because of the relative velocity difference. The difference is partly rooted in moving in the same or opposite directions.

In summary, Fig. 1a–c suggests that e-scooter-to-pedestrian interaction on sidewalks narrower than 2.0 m falls in the danger zone. Furthermore, the objective safety is the lowest when the sidewalk width is 1.6 m. In addition, the e-scooter-to-e-scooter interaction has low $T_p$ in all experiment widths and is the lowest at 3.0 m. Moreover, Fig. 1c indicates that $T_p$ and objective safety are much lower during the facing case.

The presence of other pedestrians affects the results. The crowd trials demonstrate to what extent the presence of others affects the $T_p$ and weakens the correlations. In other words, we compare the results between single and multiple pedestrian trials to show the effect of presence of others on $T_p$ and its correlation with the reported discomfort in subjective safety estimation.

To evaluate the generalizability of the observations made during the isolated trials, we perform interaction setting E, where an e-scooter passes through an arbitrary crowd, facing some pedestrians and overtaking the rest. The pedestrians with facing and overtaking interaction formats are accordingly categorized and compared in different figures. We perform the crowd trials on sidewalks with 2.0 and 3.0 m widths and report the $T_p$. However, since we do not perform the overtaking procedure for the width of 2.0 m among the isolated trials, we can only discuss the $T_p$ for the width of 3.0 m in the overtaking format.

Figure 1d shows the $T_p$ for facing cases during isolated and crowd experiments on sidewalks with widths of 2.0 and 3.0 m. The insignificant difference in $T_p$ between the crowd trials on sidewalks with different widths highlights that the pedestrian density is so high that the difference in width does not matter anymore. Thus, the $T_p$ is not as

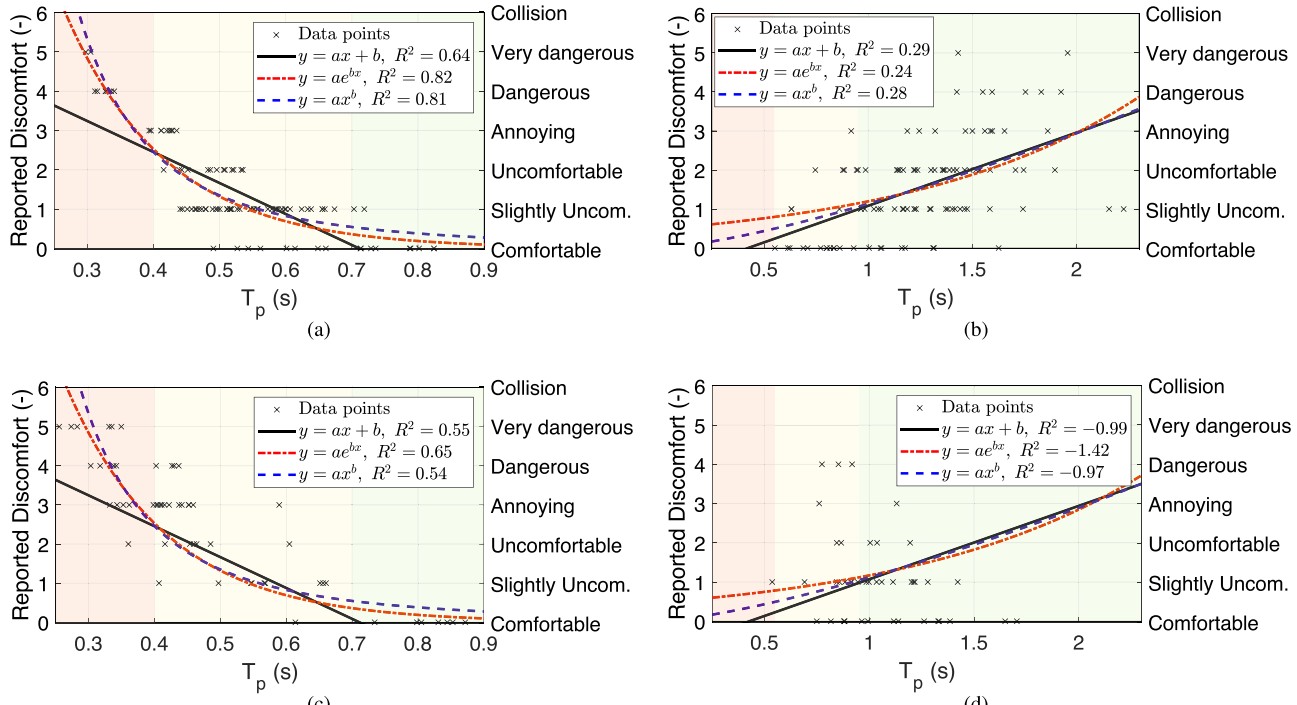

**Fig. 2 | $T_p$ and pedestrian discomfort correlation.** Fitted functions to estimate the pedestrian discomfort using $T_p$ for **a** isolated trials when a pedestrian faces an e-scooter, interaction setting A; **b** isolated trials when an e-scooter overtakes a pedestrian, interaction setting B; **c** crowd trials when a pedestrian faces an e-scooter, interaction setting E; **d** crowd trials when an e-scooter overtakes a pedestrian, interaction setting E; Table 1 presents the coefficients of determination and the constants. Data points from crowd trials, the interaction setting E, are separated according to the interaction format, i.e., facing or overtaking.

statistically different as it is between the isolated trials where the extra space is actually available.

The higher pedestrian density during the crowd trials on a sidewalk with a 3.0 m width dominates the e-scooter's movement patterns because it limits manoeuvrability. Therefore, $T_p$ is lower during the crowd trials than during the isolated trials. We expect the same pattern, i.e., lower $T_p$ when multiple pedestrians are involved, on a sidewalk with a 2.0 m width. But, the e-scooter moved more dangerously during the isolated trials than during the crowd trials on the narrower sidewalk. Therefore, comparing the $T_p$ over the mentioned cases further supports the dangerous behaviour of the e-scooter riders on sidewalk widths around 1.6–2.0 m due to the overestimation of the extra space. Moreover, the results align with patterns of a previously introduced safety metric for pedestrians; Liu et al.[14] presented the force index behaviour using Monte Carlo simulations, showing an optimal width for shared sidewalks with e-scooters.

Figure 1e is the $T_p$ for overtaking cases during isolated and crowd experiments on a sidewalk with a width of 3.0 m. Although the crowd and isolated trials have almost the same range, the high pedestrian density leads to lower $T_p$. Moreover, comparing Fig. 1d and e reveals that the difference in $T_p$ between facing and overtaking cases is also present on sidewalks with high pedestrian densities, further supporting the corresponding discussion on the interaction format effect.

**Subjective safety estimation**

We study the relationship between the reported discomfort or subjective safety and $T_p$. By fitting curves to the collected data from interaction settings A and B, we compare their $R^2$, the goodness of fit. Briefly, $R^2$, or the coefficient of determination, evaluates how well a curve fits a set of actual data points; if the data mean-value is used as the predictor, $R^2 = 0$, and for a perfect predictor, $R^2 = 1$. Thus, $R^2$ shows how much a fit is better than simple averaging. For more information, see ref. 30.

We show a strong correlation between subjective safety and $T_p$ when pedestrians see the e-scooter. However, when they don't, there is no relation, as expected. Although the PTTC is bilateral between the e-scooter rider and the pedestrian, the rider's reported subjective safety does not correlate with $T_p$. An explanation is that other parameters, for example, balancing the e-scooter or movement restrictions due to non-holonomic constraints, affect the rider's discomfort and blur the $T_p$ role.

Figure 2a, b show the data points collected from the pedestrians during the interaction settings A and B, respectively. For each interaction format, we fit functions with two constants per function for a fair evaluation. The constants may vary with extraneous variables, including but not limited to demographics, cultural differences, and user experience. The functions are nonlinear because the relation between safety metrics and avoidance manoeuvres depends on speed (see ref. 31). The candidate functions must satisfy two observed human behaviours. First, we do not feel any discomfort when the $T_p$ is long enough. On the other hand, when the $T_p$ is short enough, we feel extreme discomfort. Therefore, the functions should converge to zero when the $T_p$ is going to infinity, and they should have significantly large values when it approaches zero. In addition to lines as a base for comparison, we select and fit exponential and power functions. Table 1 presents the functions, their coefficient of determination or $R^2$, and the obtained constants for each case.

The $R^2$ values for the facing trials' fits in Fig. 2a show a statistically significant relationship between the reported discomfort and $T_p$. Considering the existing randomness in human behaviour, the goodness of fit for both estimators is significant. Moreover, for the same reason, the difference between the fitted curves is not meaningful and possibly accidental, and therefore, both candidates are suitable for discomfort estimations.

However, in Fig. 2b, where the pedestrian can not see the e-scooter approaching from behind, the relation is insignificant; $R^2 \leq 0.3$ generally indicates none or a weak correlation. We expected

**Table 1 | $R^2$ as the goodness of fit's measure and obtained constants for the fitted curves in Fig. 2**

| Interaction format | Fit type | Equation | a | b | $R^2$—Isolated trials | $R^2$—Crowd trials |
|---|---|---|---|---|---|---|
| Facing | Line | $y = ax + b$ | −7.9 | 5.60 | 0.64; Fig. 2a | 0.55; Fig. 2c |
| Facing | Exponential | $y = ae^{bx}$ | 33.9 | −6.50 | 0.82; Fig. 2a | 0.65; Fig. 2c |
| Facing | Power | $y = ax^b$ | 0.21 | −2.70 | 0.81; Fig. 2a | 0.54; Fig. 2c |
| Overtaking | Line | $y = ax + b$ | 1.88 | −0.79 | 0.29; Fig. 2b | −0.99; Fig. 2d |
| Overtaking | Exponential | $y = ae^{bx}$ | 0.49 | 0.90 | 0.24; Fig. 2b | −1.41; Fig. 2d |
| Overtaking | Power | $y = ax^b$ | 1.13 | 1.38 | 0.28; Fig. 2b | −0.97; Fig. 2d |

Fig. 2a, b are the isolated trials and Fig. 2c, d are the crowd trials.

this because the pedestrian can not see the e-scooter and does not understand the PTTC and, therefore, can not report correlated discomforts. The same argument applies to relative distance and velocity at the critical instance. Moreover, we did not find any meaningful relationship between the reported discomfort and lateral distance or relative lateral velocity. We believe the sudden appearance of the e-scooter beside the pedestrian is the primary cause of the reported discomfort. The randomness in the subjects' interpretation of safety dilutes the impact of other variables. On a separate note, the lateral distance and speed are mildly correlated on narrow sidewalks, but the correlation disappears on wider sidewalks. In facing trials, the correlation $R^2$s are 0.51, 0.50, 0.05, 0.00, and 0.10 for widths 1.3, 1.6, 2.0, 2.4, and 3.0 m, respectively; In the passing trial with 3.0 m width, $R^2 = 0.25$.

To evaluate the performances of the fitted functions in Table 1, we use them to estimate pedestrians' subjective safety during a multiple pedestrian setup, interaction setting E. Since the reported discomfort during the isolated trials, i.e., Fig. 2a and b, are significantly different for different interaction formats, we separate the pedestrians' reported safety based on their movement directions. Nonetheless, we do not categorize the data based on sidewalk width, even though half the trials are performed on a width of 2.0 m and the other half on a width of 3.0 m. Figure 2c presents the reported discomfort of the pedestrians facing the e-scooter and Fig. 2d has the reported discomfort of the pedestrians being overtaken by the e-scooter collected using the questionnaires. In addition, the previously fitted functions and their goodness of fit to the crowd trials are also presented.

The results show a moderate to strong correlation between the fitted functions of $T_p$ and the subjective safety when the e-scooter faces the pedestrian; $R^2 = 0.65$ for the exponential function, $R^2 = 0.54$ for the power function, and $R^2 = 0.55$ for the line. Although the exponential function performs better than the power function, the goodness of fit difference is insufficient to say it better estimates subjective safety confidently. However, if $T_p < 0.3$ s are excluded, $R^2$ is 0.61, 0.6, and 0.52 for the exponential, power, and line functions, respectively. Thus, when $T_p < 0.3$ s, the power function estimations are poor, and the advantage of the exponential function happens here. Since $T_p < 0.3$ is an uncomfortable area for pedestrians, the slight advantage happens in a critical span. Therefore, we conclude that the exponential function better fits the estimation requirements.

In contrast, there is no correlation between $T_p$ and reported discomfort when the e-scooter approaches the pedestrian from behind. The goodness of fit demonstrates that the weak correlation observed during the isolated trials is not present during the crowd trials; $R^2 = −1.42$ for the exponential function, $R^2 = −0.97$ for the power function, and $R^2 = −0.99$ for the line. Thus, we speculate that the main contributor to the reported discomfort is the sudden appearance of the e-scooter beside the pedestrian. The variation in the reported discomfort is due to the randomness in pedestrians' evaluation of subjective safety.

In addition, the pedestrians' reported discomfort difference between the facing and the overtaking trials is not statistically significant. On a sidewalk with a 3.0 m width for the facing trials, Fig. 2a, the average reported discomfort is 1.47 with a standard deviation of

1.24 compared to the overtaking trials, Fig. 2b, with an average of 1.57 and a standard deviation of 1.22. A two-sample $t$-test fails to reject the null hypothesis with $p = 0.57$. The same trials in Fig. 1c demonstrate higher $T_p$ for the overtaking case. While the high $T_p$ corresponds to low reported discomfort in the facing scenario, the overtaking trials do not follow the pattern. The reported discomfort correlates with the $T_p$ in facing trials. In overtaking cases, it does not correlate with $T_p$, suggesting other contributing factors dominate the reported discomfort.

In conclusion, the crowd trial findings support the generalizability of the observation during the isolated trials, like the drop in $T_p$ and lower objective safety in sidewalks with moderate width. In addition, the fitted functions estimation of the pedestrian subjective safety using the $T_p$ during facing trials have moderate to high $R^2$. During the overtaking trials, no verifiable relation is observed.

## Discussion

Dozza et al.[13] tested the braking abilities of e-scooters and concluded that relying only on braking does not ensure safe interaction with other objects, whereas steering away may be a better solution. However, this study evaluates combined braking and steering away in e-scooters, revealing that the available response time is not short enough with current riders' behaviour to ensure safety in the sense of PTTC. Our findings contribute to three riders' behaviour-modifying approaches: policy-making, proper urban design, and technological upgrades. In addition, the results provide insights into integrating mobile robots into our sidewalks.

Imposing policies, for instance, regulating e-scooters, mandatory training, and imposing licenses contribute to safer interaction on sidewalks, especially noting that over half of the e-scooter accidents happen during the first 10 rides; see Austin Public Health report[32]. Currently, $T_p$ is too low for most cases for safe interaction, and there is not enough reaction time even when the pedestrians see the rider and the avoidance is cooperative. Thus, the policies should focus on increasing $T_p$. The results provide margins to test policies through simulations, design proper training instructions, and evaluate metrics for licensing, among others.

In addition, the experiments show that e-scooters on shared narrow sidewalks with lower pedestrian density are more dangerous for pedestrians than on broad sidewalks with higher density. The presence of multiple e-scooters escalates the issue. Specifically, with current riding behaviour patterns, sidewalks under 2.0 m in width show low $T_p$ and very close misses. Since maintaining a lower bound for PTTC is a sufficient condition for collision avoidance and not a necessary one, the near-misses, although causing pedestrian discomfort, may actually be safe. However, a rider behaviour modifying action, keeping PTTC above a certain level, guarantees pedestrian safety. Moreover, it improves pedestrian comfort and subjective safety in a shared space.

An urban designer may also apply $T_p$ to estimate pedestrians' subjective safety and, using simulation platforms, pre-evaluate pedestrians' comfort in future shared public spaces. For example, considering the available space, they can decide whether to add a separate lane for e-scooters and the optimal width of that lane.

Technological upgrades can also modify riders' behaviour. Setting a lower limit for PTTC provides a sufficient condition for pedestrian safety since the rider has enough time to react in the worst-case situation, i.e., moving directly toward the pedestrian. The drawback is that the sufficient condition may be conservative in some interactions. The road traffic is structured and regulated. In addition, drivers are trained and licensed, which causes very low $T_p$ in safe interactions, making the PTTC application impractical. However, on shared sidewalks without e-scooter lanes, PTTC serves as a metric to quantify how near a miss is, and controlling it guarantees pedestrians' safety and increases their comfort. Examples of technological upgrades are warning systems and active interventions; warning systems notify the rider of low PTTC, while active interventions engage by braking or steering.

Warning systems measure PTTC in real-time, compare it to preset values, and warn the rider if driving aggressively. The alarm, for example, a sound alarm, a vibration, or a light indicator, encourages the rider to increase the PTTC and $T_p$. Active interventions like braking or steering away in case of low PTTC are also an option, requiring addressing challenges like sensing issues, pedestrian heading prediction, and maintaining balance.

In summary, with the current habits of e-scooter riders and the available response time, the combined braking and steering away results in insufficient PTTC to ensure safety; the interaction might be safe with low PTTC, but high PTTC guarantees it. Moreover, it improves pedestrian subjective safety. The low PTTC issue aggravates during long trips, which are associated with high-risk behaviour (see ref. 33). We suggest modifying movement patterns through policies, training, and technological upgrades. Additionally, the study highlights the need for compatible urban design to ensure safe interaction between e-scooters and pedestrians. Furthermore, our ongoing research focuses on the imminent integration of mobile robots into public spaces, where the fitted functions are viable candidates to extend the pedestrian subjective safety estimations to mobile robots interacting with people.

## Methods

We describe PTTC and $T_p$ and compare them to TCA. We explain the procedures used in the study, the participants involved, the data collection process, the number of trials, the equipment, and the sidewalk implementation. The Institutional Review Board (IRB) approved all methods mentioned in this section. In addition, we introduce the recorded variables and report their descriptive statistics for each interaction setting while mentioning possible limiting factors.

### Projected time-to-collision

For years, TTC has been widely used as a safety metric on roads where there are strict traffic laws and the vehicles' movement patterns are mainly 1-dimensional (see ref. 34). However, due to the lack of traffic laws, pedestrians' behaviour on sidewalks is unstructured, and the users' movement patterns are majorly 2-dimensional. We extend TTC to 2-dimensional space and call it projected time-to-collision (PTTC) since the extension roots in the pedestrian perception of approaching agents, and we formulate it by projecting TTC on the line of sight.

TTC is the remaining time to an impact between two approaching objects on a straight line assuming constant velocities and centred masses. When the two agents, for example, a pedestrian and an e-scooter, interact, their behaviour depends on how they perceive each other rather than how they move. In a 2-dimensional space, the two approaching agents may never theoretically collide because their trajectories are skew lines. However, their behaviour is affected by the rate they seemingly approach each other and perceive a potential collision. Fig. 3 is a shared space with two agents modelled as centred masses, e.g., an e-scooter and a pedestrian. $\mathbf{p}_i$, $\mathbf{p}_j$, $\mathbf{v}_i$, and $\mathbf{v}_j$ are their positions and velocity vectors in the local coordinate frame $x$–$y$,

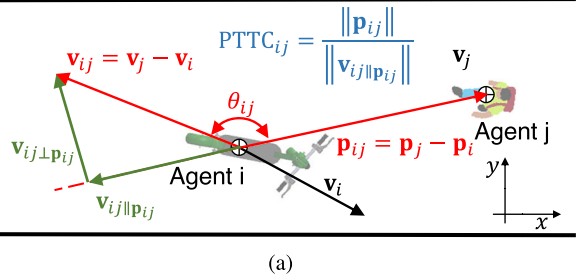

(a)

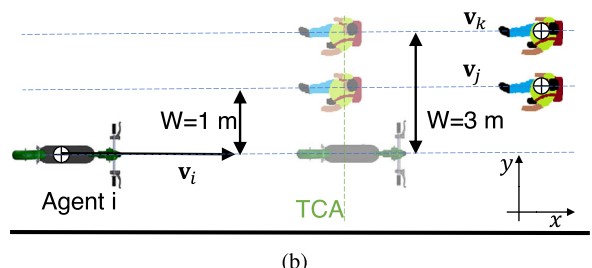

(b)

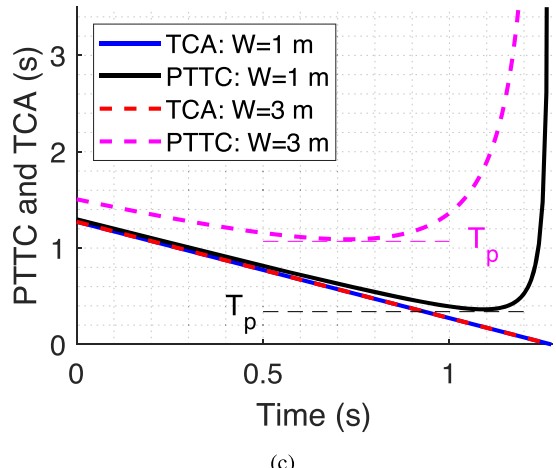

(c)

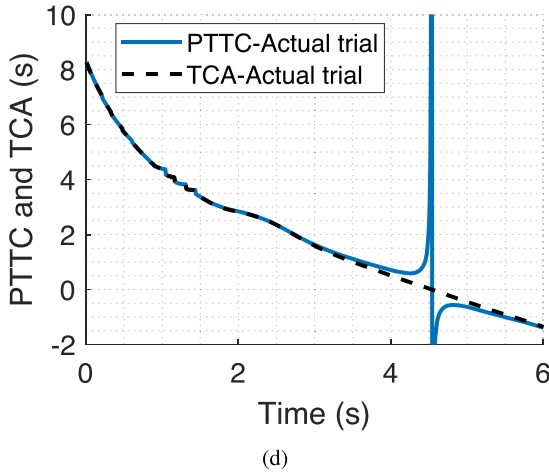

(d)

**Fig. 3 | PTTC and TCA introduction and evolution. a** The perceived approach rate $\mathbf{v}_{ij\|\mathbf{p}_{ij}}$ and PTTC definitions for agents modelled as centred masses. **b** illustration of the time to the closest approach (TCA) when an e-scooter $i$ faces a pedestrian at $W = 1$ m and $W = 3$ m. PTTC and TCA change over time in **c** the hypothetical situation and **d** an actual trial.

respectively; subscripts $i$ and $j$ denote e-scooter and pedestrian. The relative position and velocity of agent $j$ with respect to agent $i$ are $\mathbf{p}_{ij} = \mathbf{p}_j - \mathbf{p}_i$ and $\mathbf{v}_{ij} = \mathbf{v}_j - \mathbf{v}_i$. The agents' perception of TTC depends on the rate at which they approach each other in the line of sight, $\frac{d}{dt}\|\mathbf{p}_{ij}\|$. Therefore, the obtained vector $\mathbf{v}_{ij\|\mathbf{p}_{ij}}$ from projecting $\mathbf{v}_{ij}$ onto $\mathbf{p}_{ij}$ is the perceived approach rate,

$$\left|\frac{d}{dt}\|\mathbf{p}_{ij}\|\right| = \|\mathbf{v}_{ij\|\mathbf{p}_{ij}}\| = \frac{|\mathbf{p}_{ij}\cdot\mathbf{v}_{ij}|}{\|\mathbf{p}_{ij}\|}, \quad (1)$$

where "$\cdot$" is the vector's inner product. Thus,

$$\text{PTTC}_{ij} = \frac{\|\mathbf{p}_{ij}\|^2}{|\mathbf{p}_{ij}\cdot\mathbf{v}_{ij}|} = \frac{\|\mathbf{p}_{ij}\|}{\|\mathbf{v}_{ij}\| \, |\cos\theta_{ij}|}, \quad (2)$$

where $\theta_{ij}$ is the angle between the relative position and velocity. PTTC is positive since negative values mean that the agents' distance is increasing and there is no safety concern.

During an interaction between two approaching agents on a sidewalk, PTTC is large and equal to TTC when they are far; $\theta \approx 0$. In addition, since $\cos(\theta)$ decreases at a slower rate than $\|\mathbf{p}_{ij}\|$, PTTC decreases. As they get closer to each other, $\cos(\theta)$ decreases faster than $\|\mathbf{p}_{ij}\|$, and therefore PTTC starts increasing, creating a minimum point in PTTC time trajectory. $T_p$ is the minimum PTTC; we use it as the most critical point in the two agents' interaction. Note that $T_p$ happens before the agents pass each other, and it is bilateral and has the same value for both interacting agents (see Fig. 3c and d). In addition, when the agents' trajectories are skewed lines, an actual collision will not happen. For such cases, $T_p$ does not predict a collision but acts as a metric to assess how near the miss is. When the $\cos\theta$ approaches zero, the agents are not moving toward each other, and therefore, there is no collision concern, although the relative distance may be minimal. In this case, the $T_p$ is large and indicates safe interaction.

### Time of closest approach

Time of closest approach (TCA) in two agents' interaction is the time it takes for the agents to minimize their relative distance assuming constant velocities. Schwarz's[29] TCA formulation, in our terms, is

$$\text{TCA} = \frac{-\mathbf{p}_{ij}\cdot\mathbf{v}_{ij}}{\|\mathbf{v}_{ij}\|^2}. \quad (3)$$

Figure 3b is a shared space similar to Fig. 3a. The agents move toward each other with constant velocities. After TCA seconds, the agents reach a point where the e-scooter–pedestrian distance is minimal. Fig. 3c is the evolution of PTTC and TCA for the two cases in Fig. 3b up to the passing point; $T_p$ is also shown. Note that the change in W does not affect TCA in the depicted scenario, and pedestrian $j$ has the same TCA as pedestrian $k$ while $T_p$ is sensitive to W. Figure 3d is the evolution of TCA and PTTC for an actual trial. PTTC has a vertical asymptote when $\cos(\theta)$ approaches zero at the passing point. However, it does not matter in safety assessments because the agents have practically passed each other, and large $T_p$s are safe.

### Design of experiments

This section introduces the interaction settings to collect the trajectories and experiment setup. In this paper, the interaction format points to the facing and overtaking situations, agents' types refer to the neighbouring agent category, i.e., pedestrian or e-scooter, and the instructed markings on the ground are virtual curbs determining the lane width. The participants were told to stay inside coloured lines marked on the ground, each corresponding to a specific width. The experiments were performed in five settings depicted in Fig. 4; the number of trials, i.e., the sample size, is denoted by $S_s$.

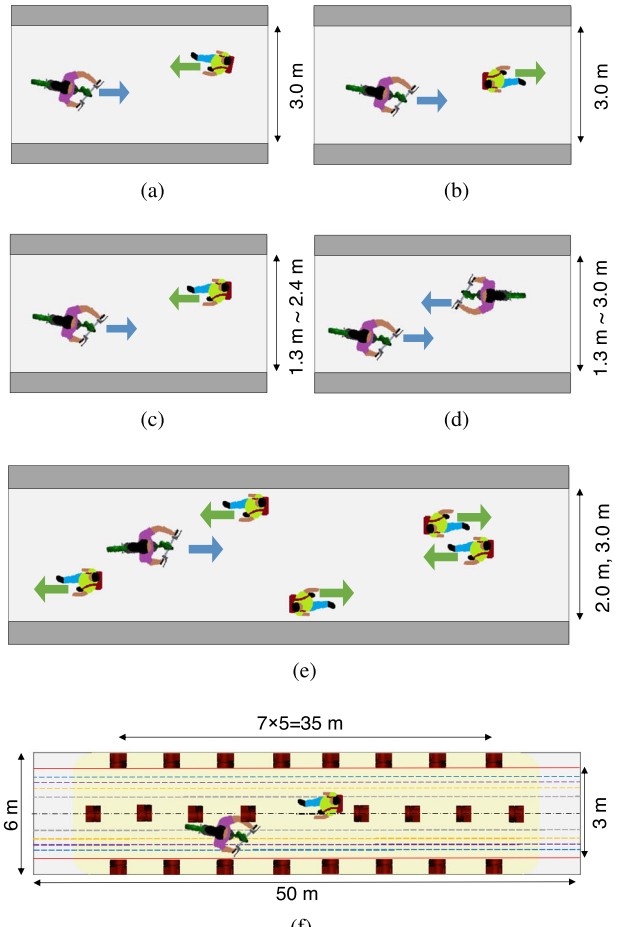

**Fig. 4 | Experiment settings and motion capture arrangement.** The experiments were performed in five settings: **a** Interaction setting A with an e-scooter and a pedestrian moving toward each other, **b** interaction setting B with an e-scooter overtaking a pedestrian, **c** interaction setting C with an e-scooter facing a pedestrian on sidewalks with different widths, **d** interaction setting D with an e-scooter moving towards another e-scooter, **e** interaction setting E with an e-scooter cruising through multiple pedestrians and **f** is the experiments' hallway and camera arrangement.

- *Interaction setting A*: Ten participants attend experiments in which an e-scooter and a pedestrian move towards each other on a 3.0 m wide sidewalk and pass by, i.e., Figs. 4a and 5a. Each pair performs ten trials. The rider and the pedestrian then swap roles and perform ten more trials, resulting in a total of 100 trials; $S_s = 100$.
- *Interaction setting B*: The e-scooter overtakes a pedestrian from behind on a 3.0 m wide sidewalk, i.e., Figs. 4b and 5b. With role swapping, the number of trials is identical to setting A, totalling 100 trials; $S_s = 100$.
- *Interaction setting C*: An e-scooter faces a pedestrian on sidewalks with different widths (1.3, 1.6, 2.0, and 2.4 m), i.e., Fig. 4c. Six participants carry out the experiments, each pair performing ten trials, then swapping roles and repeating the procedure for the next width, resulting in a total of 240 runs; $S_s = 240$.
- *Interaction setting D*: Six participants conduct experiments where one e-scooter faces another within sidewalk widths of 1.3, 2.0, and 3.0 m, i.e., Fig. 4d. Each pair performs ten trials. Although the interaction is symmetric, we did not count each trial twice to keep the data sample independent. The process results in 90 trials; $S_s = 90$.
- *Interaction setting E*: An e-scooter passes through a hallway with widths 2.0 and 3.0 m where five pedestrians are arbitrarily

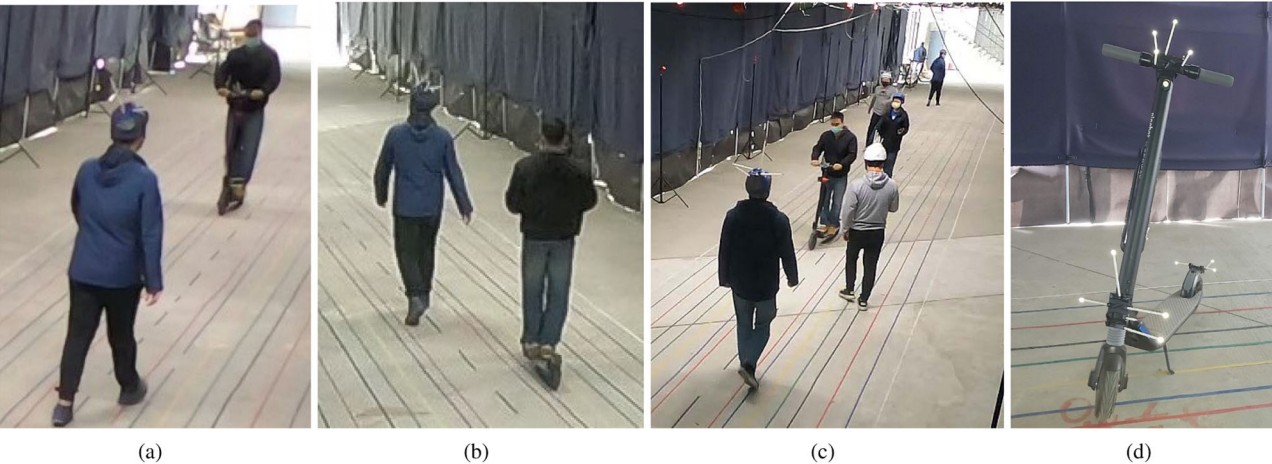

**Fig. 5 | Experiment trials. a** An e-scooter faces a pedestrian. **b** An e-scooter overtakes a pedestrian. **c** An e-scooter moves through a pedestrian crowd replicating an actual sidewalk. **d** The retro-reflective markers installed on the e-scooter.

## Table 2 | Experiment design summary

| Setting | Format | Type | Sample size | Width (m) |
|---------|--------|------|-------------|-----------|
| A | Facing | Pedestrian | 100 | 3.0 |
| B | Overtaking | Pedestrian | 100 | 3.0 |
| C | Facing | Pedestrian | 240 | 1.3/1.6/2.0/2.4 |
| D | Facing | E-scooter | 90 | 1.3/2.0/3.0 |
| E | Mixed | Pedestrians | 20 | 2.0/3.0 |

positioned and moving towards different ends of the hallway, i.e., Figs. 4e and 5c. The experiment is conducted twenty times, 10 for each width; $S_s = 20$.

Table 2 summarizes the design of the experiments.

After each trial, the participants complete a questionnaire reporting their maximum discomfort level. We use 200 data points from pedestrians involved in interaction settings A and B for curve fitting, $2 \times 100$, and 100 data points from pedestrians involved in interaction setting E for validation, $5 \times 20$. The subjects are instructed to report their discomfort using 0-Comfortable, 1-Slightly uncomfortable, 2-Uncomfortable, 3-Annoying, 4-Dangerous, 5-Very dangerous or near crash, and 6-Collision. The description for each number reduces the personal interpretation of the discomfort levels.

### Experiment setup

All participants are 20–35-year-old university students and are novice e-scooter riders. Some have a couple of rides before, and some have none. Two participants have a few months of experience, and no participant is a regular e-scooter user; age and experience may affect the e-scooter rider's behaviour as an extraneous variable. Thus, our results are verified for the sample. They may directly apply to the whole e-scooter user population or require re-evaluation for generalization. Three out of ten participants are female; however, we perform no sex and gender-based analyses due to a small sample population and the generalization limitation. Before the trials, the participants ride around the experiment area and get familiar with the environment. They know that the research goal is to study e-scooter riders' behaviour in general. However, they are unaware of specific research goals. The experiments were performed on the National Cheng Kung University (NCKU) campus from October 2021 to January 2022, spanning a duration of 4 months. All participants provided informed consent prior to their participation in the study, and all data were collected under ethical principles.

The e-scooter used in the experiments is an ES4 Ninebot Kick scooter from Segway. The model's maximum velocity is 8.3 m/s, yet, out of concern for participants, a top speed of 5 m/s is forced in the e-scooter's internal setting. In addition, we instruct the rider and the pedestrians to move at a comfortable speed.

We install multiple groups of retro-reflective markers on the pedestrians' helmets (Fig. 5a–c) and on the scooter (Fig. 5d). The markers in each group form a unique 3D configuration, enabling the motion capture system to locate their geometric center as the agents' relative position to a predefined origin. Therefore, the participants' head rotation does not significantly affect the position as long as the geometric center does not change. However, the head moves sideways with each step as the pedestrian walks. We did not average/compensate for the lateral movement since our method is based on pedestrian line of sight changing with the head's instantaneous position. As arranged in Fig. 5f, a motion capture system with 24 infrared cameras (Optitrack Flex 13) sees the markers' reflection and locates the participants. Sixteen cameras installed on the hallway's walls cover the central area. The eight cameras on the ceiling provide redundancy in marker detection to compensate for blind spots and partial blockage of the cameras' line of sight by participants' bodies.

The hallway dimensions are 6 m × 50 m, but we ask the participants to stay inside a predefined width marked on the ground as virtual sidewalk curbs. Moreover, the cameras target the central 35 m for data collection. The rest of the hallway is for acceleration and deceleration. For the reported discomfort evaluations, we select the widths 2.0 and 3.0 m because they are not too wide to let the participants pass each other without any significant interaction and are not too narrow to affect the participants' feelings and obscure their judgement of discomfort from the e-scooter.

The data-processing software (Motive) records the timed trajectories of the participants at 120 Hz. The recorded timed trajectories are used to calculate relative positions and velocities. We calculate PTTC using these relative states at each moment and identify the minimum $T_p$. Supplementary movie 1 clarifies the procedure.

After conducting the experiments, we calculate the PTTC and its minimum, $T_p$, using $\mathbf{p}_{ij}$ and $\mathbf{v}_{ij}$ during each trial. Additionally, we report the relative velocity and position at $T_p$'s occurrence moment as critical distance and speed. We also note the relative lateral distance and velocity of the agents at the passing point. To better understand the data distribution, Table 3 presents the descriptive statistics, i.e., the mean, the median, and the standard deviation for each interaction setting per width. We provide the collected information in Source Data.

**Table 3 | Summary of the calculated variables using the collected trajectories presented by the descriptive statistics, i.e., the mean, the median, and the standard deviation (as a percentage of the mean) for each interaction setting per width**

| Interaction setting (–) | Width (m) | Sample size (–) | $T_p$ (s) | Relative position (m) | Relative velocity (m/s) | Lateral distance (m) | Lateral velocity (m/s) |
|---|---|---|---|---|---|---|---|
| A | 3.0 | 100 | (0.53,0.52,24) | (1.82,1.77,23) | (5.10,4.97,16) | (1.29,1.26,22) | (5.02,4.91,16) |
| B | 3.0 | 100 | (1.26,1.24,28) | (1.78,1.70,29) | (2.18,2.28,26) | (1.27,1.23,25) | (2.29,2.38,26) |
| C | 1.3 | 60 | (0.37,0.37,11) | (1.21,1.19,16) | (4.66,4.57,12) | (0.85,0.82,16) | (4.66,4.59,12) |
| C | 1.6 | 60 | (0.32,0.30,30) | (1.30,1.29,19) | (5.93,6.13,12) | (0.92,0.92,20) | (5.95,6.18,12) |
| C | 2.0 | 60 | (0.42,0.39,23) | (1.54,1.44,20) | (5.42,5.40,12) | (1.09,1.03,19) | (5.41,5.38,13) |
| C | 2.4 | 60 | (0.49,0.49,27) | (1.57,1.47,23) | (4.70,4.41,19) | (1.11,1.01,22) | (4.71,4.45,19) |
| D | 1.3 | 30 | (0.25,0.25,10) | (1.18,1.19,8) | (6.88,6.92,10) | (0.84,0.84,8) | (6.93,6.99,10) |
| D | 2.0 | 30 | (0.29,0.28,13) | (1.77,1.88,21) | (8.86,9.47,13) | (1.26,1.34,21) | (8.95,9.55,13) |
| D | 3.0 | 30 | (0.24,0.22,22) | (1.41,1.41,15) | (8.41,8.63,7) | (1.00,1.01,15) | (8.45,8.6,8) |

Interaction setting *E* is not presented because it has a mixed crowd and the unsorted collected data may be misleading. For the collected data, see Source Data.

## Limitations

A few factors may limit the statistical significance and generalizability of the results. We use the maximum velocity limitation of 5 m/s out of safety concerns during the experiments. However, this limitation may affect some cases. For instance, in interaction setting D at a width of 2.0 m, when two scooters are approaching each other, the median value of the lateral velocity is 9.55 m/s, while the mean is 8.95 m/s. The upper bound of 10 m/s (5 m/s for each scooter in facing settings) for the interaction causes the distribution to deviate from normal and become asymmetric. Nonetheless, we still use the data in our research, considering that limiting the maximum velocity may apply to many users on sidewalks.

Moreover, e-scooter riders have balancing issues at low speeds imposing a lower limit on their movement patterns. Most users hesitate to move slower than ~2 m/s (absolute observed minimum of 1.4 m/s) because maintaining the balance takes too much effort. Interaction settings in Table 3 are not subject to this lower bound because the e-scooter velocity is far from the minimum. However, for the crowd trials, when the pedestrian density is high, i.e., interaction setting E, this may affect the results making the distributions asymmetric.

Statistical significance tests assume the normality of the sample distribution. The statistical normality tests confirm the normality of the samples for most cases; see Supplementary Table 1. However, in some scenarios, the data are not normal, perhaps because of the forced top speed limit for safety and the natural minimum speed limit to maintain the balance. The deviation from normality limits the credibility of the significance tests.

Moreover, presenting curbs/walls as marked lines on the experiment hallway floor affects the rider's behaviour. We believe the riders keep more distance from curbs/walls than lines on the ground, getting closer to the pedestrians and resulting in lower $T_p$. In addition, the participants during the experiment are vigilant, while on actual walkways, may be inattentive or distracted and likely to behave more dangerously. Furthermore, repeated trials and switching roles can impact riding behaviour through experience accumulation, too.

While this study provides insights into pedestrians' safety and comfort when interacting with e-scooters, the sample size may limit the generalizability of the results and not capture the full spectrum of variability in the broader population. Larger sample sizes would be beneficial for obtaining more robust and representative results, allowing for greater confidence in the generalizability of the findings to the target population.

Other extraneous variables like participants' experience, demographics, presence of other users, and e-scooter mechanical abilities limit the generalizability of the results. Further experiments with a population similar to e-scooter users and more realistic situations may relax the limitations.

## Reporting summary

Further information on research design is available in the Nature Portfolio Reporting Summary linked to this article.

## Data availability

All data generated or analysed during this study are included in this published article and its supplementary information files, and also are available from the corresponding author upon request. Source data are provided as a Source Data file. Source data are provided with this paper.

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

## Acknowledgements

This work was supported in part by the National Science and Technology Council (NSTC), Taiwan, under Grant NSTC 112-2636-E-006-001 and NSTC 112-2628-E-006-014-MY3. Moreover, the study received IRB approval from National Cheng Kung University (Approval No. NCKU HREC-E-109-356-2) on October 23, 2020, which was subsequently extended to January 31, 2022.

## Author contributions

A.J. conceptualized, designed, and performed the experiments, collected and analysed the data, and wrote the original draft. Y.-C.L. conceptualized, provided the resources and the funding, supervised the experiments, reviewed the analysis, and edited the final draft.

## Competing interests

The authors declare no competing interests.
