## [Peer Review File · Nature Communications]

Pedestrians' safety using projected time-to-collision to electric scootersREVIEWER COMMENTS

Reviewer #1 (Remarks to the Author):

This paper describes a field experiment in which ten students collected data from a motion capture system while interacting as pedestrians and e-scooterists in different conditions. The experimental set-up with 24 cameras covering an area of 6 m by 50 m is unprecedented for this type of application. The video attached to the submission helped explain the protocol. Several major concerns keep me from recommending the publication of this paper in its current format, especially in a high-profile journal such as Nature Communications. Below, I list my major concerns.

1. Some of the main arguments and conclusions in the article are not supported by the results.

a. The author calculated an extension of TTC, PTTC, and used this metric to estimate T_p , the minimum value of PTTC in the (longitudinal) interactions tested in their experiment. While at long distances TTC and PTTC may be roughly the same, as the two road users get closer, these two metrics may be very different. The authors explain the rationale for using PTTC and they also (experimentally) show that there is a relation between T_p and perceived safety. Several surrogate safety measures, similar to TTC, have been suggested before (e.g., time to closest approach; TTCA; Schwartz 2015). How does PTTC compare to TTCA? This is important for two reasons. 1) looking at eq. 2, it looks as if θ_{ij} may cause the denominator to be close to 0, making PTTC very large when the two road users may be closest, and 2) collisions may only happen if the two road users are on a collision path, so T_p is not necessarily predicting a collision. Showing how PTTC changes over time as two road users approach each other may help the reader appreciate 1) when T_p actually happen, 2) what is the effect of steering on PTTC and T_p , 3) how heading data need to be filtered for PTTC to be stable.

The authors use T_p to claim: "our experiments show that, with current riders' movement patterns and on narrow sidewalks, even combined braking-steering-based collision avoidances are dangerous." I am not convinced the results prove this. Similarly, the authors conclude "even the combined braking and steering away is insufficient to ensure safety". This statement is also not proven by the results and, most likely wrong, since steering was the maneuver that kept the participants safe during the experiment.

In longitudinal interactions, such as the ones explored in this paper, steering is the most common evasive maneuver. (Please notice that the relation between safety metrics and avoidance maneuvers such as braking and steering is not linear and changes depending on speed; Brännström 2010.) The authors anchor their results to response time for braking (Figure 2). While this makes sense because response time may be similar for steering, the direct comparison of T_p to reaction time is misleading. T_p may be small and if the road users are not on a collision path, they are likely safe, and no avoidance maneuver may be necessary. Further, in normal traffic, T_p for motorized vehicles meeting oncoming traffic may be extremely small (and smaller than the T_p recorded in this experiment because speeds may be 6-8 times higher), but that may not mean that the interaction is dangerous (or that we can trust T_p for ADAS such as AEB; I will come back to this point). Therefore, statements like "However, this study evaluates combined braking and steering away in e-scooters, revealing that the available response time is not short enough with current riders' behaviour, and the pedestrians face a significant risk" are hard to understand as T_p and response time are two different things.

The authors also claim: "In addition, the e-scooter-to-e-scooter interaction is dangerous in all experiment widths", then they add: "Specifically, with current riding behaviour patterns, sidewalks under 2.0 m in width are not recommended for e-scooters and require reconsideration." This might be correct, but the authors cannot make this claim based on their results, not only because of the safety metric they used but also because of the nature of the experimental set-up (I will come back to this in the next points when addressing ecological validity and the limitation of the study.)

On page 7, the paper reports "A central controller uses measured relative positions and velocities, monitors pedestrians' subjective safety, and intervenes to maintain a safe T_p , resulting in more pleasant interactions for pedestrians." If commercial AEB would act on T_p ,

the number of false positive would be unacceptable. Further, active safety systems rely on predictions. On a time-horizon of 1-2 s, autonomous interventions based on predicted T_p may not be appropriate. T_p might be a good indicator for traffic simulations or used to give warnings when an e-scooterist consistently exhibits T_p below average; however, autonomous interventions based on T_p do not seem viable. Further, pedestrian heading is very hard to measure and can change instantly, creating some severe challenges for sensing and prediction, independently of the technology available. (BTW: passive ADAS sounds like an oxymoron and may require some reference for reader to understand what you mean.)

2. The sample size of the dataset and the statistical analyses are not appropriate for the analyses and conclusions that the authors want to draw.

A total of ten participants took part in the experiment; however, for some conditions only 6 participants provided data in pairs. In practice, in some conditions, one single subject was able to influence 1/3 of the whole dataset. Taking Figure 2B as an example, some of the box plots are made of 60 points (only 30 of them unique) of which a single participant may influence 20 (directly or indirectly). As a consequence, the confidence intervals are more affected by repetitions from the same subject than by the variability across subjects. So, how should we interpret these confidence intervals? What is the effect of counting data twice (setting D) on the confidence intervals? These points are important because the authors do not provide statistical analyses but propose to visually inspect the box plots to infer statistical significance based on the overlap of their notches. Because of human variability, a larger number of participants would be necessary to make sure the results are statistically sound. Performing statistical analyses and checking that the dataset responds to the requirements of the selected statistical test may also grant more trustworthy results and offer an opportunity to control for multiple comparisons. Limiting the number of conditions and increasing instead the number of participants may help provide more value to the reader.

3. Some aspects of the methodology are confusing and need clarification.

The use and calculation of PTTC may be further backed-up. Comparing PTTC to TTCA (see above) and other surrogate safety metrics may help. In addition, showing how PTTC changes over time as the road users meet may help the reader appreciate the advantages (and disadvantages) of this metric. Further, the process for extracting trajectories from stereophotogrammetry should be explained. Here, one of the critical aspects is the heading estimation; especially for pedestrians that may walk in one direction while rotating their head in another. (BTW: typically, in Nature Communications, availability of code and data are addressed.)

Verification trials puzzle me a bit. I understand the concerns of the authors about verifying their metrics in a more realistic situation than the "single-encounter ones" used in the main protocol; however, I wonder what these verification trials verified. The position and behavior of the five pedestrians seems crucial in this situation and hard to control. I am not sure that these trials add much to the paper, because they raise questions that hard to answer; for instance: "How did you randomly position the five people?" and "How does this condition compare to reality?".

4. The limits of the study should be at least mentioned.

Like all studies performed in controlled environments and limiting the experimental protocol for safety concerns, this study also has limitations that influence the ecological validity of the results. Such limitations should be mentioned and claims such as "realistic sidewalks" should be avoided. (It is hard to defend that a painted line on tarmac is a realistic sidewalk.) Although still artificial, your own "verification trials" hint already that road-user behavior will be different than in the experimental conditions as soon as other road-users will be around. Indeed, the Discussion section would benefit from discussing the results in the light of previous literature and addressing the limitations of the study, (while limiting some of the claims about the applicability of the results that are too stretched).

5. The paper should be reorganized to help the reader understand and digest its content.

For instance, results, methods, and discussion are mixed, making it hard for the reader to find the content she is looking for.

As an example, the first part of the Results section is about methods. Further some results are interpreted and mixed with anecdotal evidence. This is an example:

"In addition, during facing interactions average pedestrian velocity at the critical point is 1.5 m/s, while during the overtaking trials, it is 1.3 m/s. Interestingly, the pedestrian moves faster to avoid the e-scooter in the line of sight, while during the overtaking trials, it doesn't have any clue of the approaching scooter and moves as usual. The difference is also partly because the e-scooter rider, accounting for the pedestrian's possible sudden movements, approaches the pedestrian more carefully during overtaking than facing trials."

Reviewer #2 (Remarks to the Author):

The growth in use of e-scooters in cities around the world and the challenge of protecting both riders and pedestrians means that the current research addresses a topic which is of widespread interest and concern.

The authors claim in the introduction that e-scooters are mainly used for first/last mile transport, connecting transport hubs to final destinations and they quote Ma et al. (2022). That article focused on this type of trip in one location (Washington DC) and so does not provide a strong basis for the claim. The authors neglect a range of other studies which demonstrate that many shared e-scooter trips are for fun/recreation, rather than first/last mile transport. In the same paragraph, the Clabaux et al. (2014) article is about motorcycles, not e-scooters and so it is unclear how it is relevant.

The term "Perceived Time-To-Collision" may be confusing for some readers. When I read the title of the manuscript, I assumed that it meant a subjective measure of TTC, analogous to the use of "perceived" in "perceived safety". The potential for confusion is exacerbated because the manuscript is comparing "PTTC" to subjective judgements of safety. I would strongly recommend changing the term. One alternative would be to keep "PTTC" but change it to stand for "Pedestrian Time-To-Collision".

Figure 2 – it would be useful to alert the reader to the discrepant scales across the five panels. Were lateral position and speed correlated?

Do you expect the relationship between PTTC and subjective safety/discomfort to be uni- or bi-directional? The subjective safety/discomfort was only for the pedestrian as far as I can gather. The comments in the results section about the lack of relationship between PTTC and discomfort suggests that the authors assume PTTC underlies discomfort.

Was level of discomfort higher in overtaking than facing?

Comparing the values obtained in the "control" scenarios with the more realistic multi-pedestrian scenario E may not be appropriate given that pedestrians may feel greater discomfort because they perceive they have less freedom to move away from the e-scooter (because of the presence of other pedestrians). On the other hand, the e-scooter rider may slow down more after passing the first pedestrian and so there is less discomfort in this situation.

Page 6, last paragraph – the authors conclude that the PTTCs are too low in most cases for safe interaction but they didn't report any collisions occurring during the collection of their experimental data. Doesn't this query this conclusion?

The extension of the PTTC to mobile robots may not be appropriate as a way to reduce pedestrian discomfort because pedestrians may use eye-contact with e-scooter riders which they could not with mobile robots (although I note the research into external HMIs on automated vehicles to improve the safety of interactions with pedestrians).

Method

In Interaction setting A and C (not clear whether this happens in B), the rider and participant swap roles. Does swapping roles influence the subjective safety because of the accumulation of

experience in the experiment?

How do participants know the sidewalk width? If there were real curbs, would this reduce subjective safety? It may have been better to conduct the experiment with physical edges to the sidewalks.

Does e-scooter always overtake pedestrian on the right? What is rule in Singapore – keep to the left or right?

Does less than 6 months of e-scooter experience mean that some participants had a small degree of e-scooter experience and some had none? Please clarify in the text.

The top speed of 5 m/s is consistent with the speed limits in countries with sidewalk speed limits for e-scooters as noted by the authors.

Response to referees-Rev. 01, NCOMMS-23-56216, “Pedestrians’ safety using **projected time-to-collision to electric scooters”**

Statement of Changes and Revisions

The authors would like to thank the reviewers for their valuable comments and suggestions. Here is a list of changes that are made according to the reviewers’ suggestions and the replies to the comments.

1. Reviewer 1’s Comment No. 1 leads to the following updates: Page 3, Right column, Line 20 to Line 27; Page 2, Left column, Line 51 to Line 54; Page 3, Right column, Line 7 to Line 9; Page 3, Right column, Line 28 to Line 45; Fig. 1(b)—(d); Page 1, Right column, Line 19 to Line 22; Page 7, Right column, Discussion, Line 26 to Line 30; Page 8, Left column, Line 47 to Line 52; Page 6, Left column, Line 13 to Line 18; Page 8, Left column, Line 23 to Line 38; Page 4, Left column, Observations, Line 40 to Line 41; Page 4, Right column, Line 36; Page 8, Left column, Line 7 to Line 16; Page 8, Left column, Line 39 to Line 46.
2. Reviewer 1’s Comment No. 2 leads to the following updates: Page 4, Right column, Line 6 to Line 8; Figure 2’s caption; Page 8, Interaction setting D; Tables 2 and 3; Fig. 2 caption; Supplementary Tables.
3. Reviewer 1’s Comment No. 3 leads to the following updates: Page 9, Right column, Line 39 to Line 48; Page 11, Data availability; Page 10, Right column, Line 1 to Line 2; Page 2, Right column, Line 21 to Line 24.
4. Reviewer 1’s Comment No. 4 leads to the following updates: Page 10, Right column, Line 3 to Line 4; Page 11, Left column, Line 5 to Line 30; Page 8, Left column, Line 43 to Line 46; Page 8, Right column, Line 2 to Line 6.
5. Reviewer 2’s Comment No. 1 leads to the following updates: Page 1, Left column, Line 3 to Line 8.
6. Reviewer 2’s Comment No. 2 leads to the following updates: the title; the abstract; Page 2, Right column, Line 14; and the Results section’s first paragraph.
7. Reviewer 2’s Comment No. 3 leads to the following updates: Fig. 2 caption; Page 6, Right column, Line 24 to Page 7, Left Column, Line 3; Page 6, Left column, Line 3 to Line 9; Page 7, Right column, Line 1 to Line 14; Page 4, Right column, Line 40 to Line 46; Page 4, Left column, Line 19 to Line 26.
8. Reviewer 2’s Comment No. 4 leads to the following updates: Page 8, Right column, Interaction setting B; Page 11, Left column, Line 20 to Line 25; Page 8, Right column,

Line 22 to Line 24; Page 11, Left column, Line 13 to Line 17; Page 9, Right column,
Line 11 to Line 14.

We further clarified issues the reviewers raised in the response letter. Please find attached the revised manuscript and a summary of the detailed responses to the reviewers.

◇ Response to Reviewer 1's Comments

The authors would like to thank the reviewer for the valuable comments and suggestions. We believe the comments lead to the improvement of this paper, especially comparing the suggested metric evolution with a previous metric and providing a detailed statistical analysis. Next, we review the reviewer's reply point-by-point (we divided the comments into subsections for easier response tracking). Moreover, the updates in the paper are highlighted and the change's location is addressed.

Comment 0- *This paper describes a field experiment in which ten students collected data from a motion capture system while interacting as pedestrians and e-scooterists in different conditions. The experimental set-up with 24 cameras covering an area of 6 m by 50 m is unprecedented for this type of application. The video attached to the submission helped explain the protocol. Several major concerns keep me from recommending the publication of this paper in its current format, especially in a high-profile journal such as Nature Communications. Below, I list my major concerns.*

Response. We are grateful for the reviewer's understanding of the extent of the experiments. We hope we can address the respected reviewer's concerns so that the outcome benefits the scientific community and the interested readers.

Comment 1 - *Some of the main arguments and conclusions in the article are not supported by the results.*

Comment 1-a. The author calculated an extension of TTC, PTTC, and used this metric to estimate T_p , the minimum value of PTTC in the (longitudinal) interactions tested in their experiment. While at long distances TTC and PTTC may be roughly the same, as the two road users get closer, these two metrics may be very different. The authors explain the rationale for using PTTC and they also (experimentally) show that there is a relation between T_p and perceived safety. Several surrogate safety measures, similar to TTC, have been suggested before (e.g., time to closest approach; TTCA; Schwartz 2015). How does PTTC compare to TTCA? This is important for two reasons. 1) looking at Eq. 2, it looks as if $\cos \theta_{ij}$ may cause the denominator to be close to 0, making PTTC very large when the two road users may be closest, and 2) collisions may only happen if the two road users are on a collision path, so T_p is not necessarily predicting a collision. Showing how PTTC changes over time as two road users approach each other may help the reader appreciate 1) when T_p actually happens, 2) what is the effect of steering on PTTC and T_p , 3) how heading data need to be filtered for PTTC to be stable.

Response. Regarding the reviewer's questions, when the $\cos \theta_{ij}$ approaches zero, the agents

(a)

Figure 1: (Fig. 1(b) in the paper) Illustration of the time to the closest approach (TCA) when an e-scooter i faces a pedestrian at $W=1$ m and $W=3$ m.

Figure 2: PTTC and TCA change over time in (a) the hypothetical situation (Fig. 1(c) in the paper); (b) an actual trial (Fig. 1(d) in the paper).

are not moving toward each other and are not on a collision path, even if the relative distance is the closest. Therefore, the T_p is large and indicates safe interaction even if the agents are close. Moreover, when the agents' trajectories are skewed lines, an actual collision will not happen. For such cases, T_p does not predict an actual collision but acts as a metric to assess how near the miss is. For example, when the $\cos \theta_{ij}$ approaches zero, the agents are not moving toward each other, and therefore, there is no collision concern, although the relative distance may be minimal. In this case, the T_p is large and indicates safe interaction.

We edit the following in the manuscript.

“Note that T_p happens before the agents pass each other, and it is bilateral and has the same value for both interacting agents; see Fig. 1(c) and (d). In addition, when the agents' trajectories are skewed lines, an actual collision will not happen. For such cases, T_p does not predict a collision but acts as a metric to assess how near the miss is. When the $\cos \theta$ approaches zero, the agents are not moving toward each other, and therefore,

there is no collision concern, although the relative distance may be minimal. In this case, the T_p is large and indicates safe interaction.”

The change affects Page 3, Right column, Line 20 to Line 27.

Additionally, we apply the reviewer’s helpful suggestions and briefly introduce TCA, comparing it to PTTC. Specifically, we compare PTTC and TCA in an oversimplified case to help the reader understand the concept. In addition, we calculate TCA and PTTC for one of our trials. We add the following to the paper to prepare the text for introduction of the Time of Closest Approach (TCA).

“In addition, Schwarz used the Time of Closest Approach (TCA), suggesting computation algorithms using bounding boxes. We compare our proposed metric to TCA.”

The change affects Page 2, Left column, Line 51 to Line 54.

“PTTC is positive since negative values mean that the agents’ distance is increasing and there is no safety concern.”

The change affects Page 3, Right column, Line 7 to Line 9.

Then, we briefly introduce TCA and compare it to PTTC.

“**Time of Closest Approach.** Time of Closest Approach (TCA) in two agents’ interaction is the time it takes for the agents to minimize their relative distance assuming constant velocities. Schwartz’s TCA formulation, in our terms, is

$$TCA = \frac{-\vec{p}_{ij} \cdot \vec{v}_{ij}}{\|\vec{v}_{ij}\|^2}. \quad (1)$$

Fig. 1(b) is a shared space similar to Fig. 1(a). The agents move toward each other with constant velocities. After TCA seconds, the agents reach a point where the e-scooter-pedestrian distance is minimal. Fig. 1(c) is the evolution of PTTC and TCA for the two cases in Fig. 1(b) up to the passing point; T_p is also shown. Note that the change in W does not affect TCA in the depicted scenario, and pedestrian j has the same TCA as pedestrian k while T_p is sensitive to W . Fig. 1(d) is the evolution of TCA and PTTC for an actual trial. PTTC has a vertical asymptote when $\cos \theta$ approaches zero at the passing point. However, it does not matter in safety assessments because the agents have practically passed each other, and large T_p s are safe. ”

The change affects Page 3, Right column, Line 28 to Line 45, and the added Fig. 1(b)–(d).

Comment 1-b. The authors use T_p to claim: “Our experiments show that, with current riders’ movement patterns and on narrow sidewalks, even combined braking-steering-based collision avoidances are dangerous.” I am not convinced the results prove this.

Similarly, the authors conclude “even the combined braking and steering away is insufficient to ensure safety”. This statement is also not proven by the results and, most likely wrong, since steering was the maneuver that kept the participants safe during the experiment. In longitudinal interactions, such as the ones explored in this paper, steering is the most common evasive maneuver. (Please notice that the relation between safety metrics and avoidance maneuvers such as braking and steering is not linear and changes depending on speed; Brännström 2010.) The authors anchor their results to response time for braking (Figure 2). While this makes sense because response time may be similar for steering, the direct comparison of T_p to reaction time is misleading. T_p may be small and if the road users are not on a collision path, they are likely safe, and no avoidance maneuver may be necessary. Further, in normal traffic, T_p for motorized vehicles meeting oncoming traffic may be extremely small (and smaller than the T_p recorded in this experiment because speeds may be 6-8 times higher), but that may not mean that the interaction is dangerous (or that we can trust T_p for ADAS such as AEB; I will come back to this point). Therefore, statements like “However, this study evaluates combined braking and steering away in e-scooters, revealing that the available response time is not short enough with current riders’ behaviour, and the pedestrians face a significant risk” are hard to understand as T_p and response time are two different things.

Response. We understand the reviewer’s point and updated the text accordingly to better match our experiments. According to the comment, we think we failed to clarify that maintaining a minimum PTTC is a sufficient condition for objective safety and not a necessary one. The metric can be a conservative condition for road traffic due to the structured driving behavior. However, on sidewalks, it guarantees pedestrian safety. We hope the following fixes resolve the issue. First, we rephrase the sentences and later in this response, we clarify the point.

“Our experiments study riders’ collision avoidance using combined braking-steering and evaluate pedestrian safety by comparing a time-to-collision variant with response time. ”

The change affects Page 1, Right column, Line 19 to Line 22.

“However, this study evaluates combined braking and steering away in e-scooters, revealing that the available response time is not short enough with current riders’ behaviour to ensure safety in the sense of PTTC.”

The change affects Page 7, Right column, Discussion, Line 26 to Line 30.

“In summary, with the current habits of e-scooter riders and the available response time, the combined braking and steering away results in insufficient PTTC to ensure safety;

the interaction might be safe with low PTTC, but high PTTC guarantees it. Moreover, it improves pedestrian subjective safety.”

The change affects Page 8, Left column, Line 47 to Line 52.

Regarding the nonlinear behavior, our results also confirm it. Assuming that the discomfort motivates the riders to evade a pedestrian, its nonlinear relation to T_p confirms that avoidance maneuvers and T_p as a safety metric are nonlinearly correlated (for example, exponentially in Fig. 3). We add the following to the manuscripts.

“For each interaction format, we fit functions with two constants per function for a fair evaluation. The constants may vary with extraneous variables, including but not limited to demographics, cultural differences, and user experience. **The functions are nonlinear because the relation between safety metrics and avoidance maneuvers depends on speed; see Brännström et al..**”

The change affects Page 6, Left column, Line 13 to Line 18.

The reviewer pointed out the difference between PTTC and the response time. Since PTTC is the time to collision in the direction of the relative position, PTTC and response time are comparable if the e-scooter is directly moving toward the pedestrian acting as TTC, the worst-case scenario. Setting a lower limit for PTTC provides a sufficient condition for pedestrian safety since the rider has enough time to react in the worst-case situation. The drawback is that the sufficient condition may be conservative in some interactions, for example, car traffic situations.

The road traffic is structured and regulated. In addition, drivers are trained and licensed, which causes a different behavior with very low T_p in safe interactions, making the PTTC application impractical. However, on shared sidewalks without e-scooter lanes, controlling PTTC, either by alarming the rider or by actively engaging to maintain the minimum, can keep pedestrians safe, although it may be a conservative approach.

In addition, PTTC serves as a metric to quantify how near a miss is. Although no collision happened during the experiments, it does not mean that the interactions are safe. Low T_p s observed during the experiments indicate that the near-misses may be dangerous.

We add the following to the paper to clarify the points in the comment.

“Technological upgrades can also modify riders’ behaviour. Setting a lower limit for PTTC provides a sufficient condition for pedestrian safety since the rider has enough time to react in the worst-case situation, i.e., moving directly toward the pedestrian. The drawback is that the sufficient condition may be conservative in some interactions. The road traffic is structured and regulated. In addition, drivers are trained and licensed,

which causes very low T_p in safe interactions, making the PTTC application impractical. However, on shared sidewalks without e-scooter lanes, PTTC serves as a metric to quantify how near a miss is, and controlling it guarantees pedestrians' safety and increases their comfort. Examples of technological upgrades are passive and active ADAS; passive ADAS notifies the rider of low PTTC, while active ADAS engages by braking or steering."

The change affects Page 8, Left column, Line 23 to Line 38.

Comment 1-c. The authors also claim: "In addition, the e-scooter-to-e-scooter interaction is dangerous in all experiment widths", then they add: "Specifically, with current riding behaviour patterns, sidewalks under 2.0 m in width are not recommended for e-scooters and require reconsideration." This might be correct, but the authors cannot make this claim based on their results, not only because of the safety metric they used but also because of the nature of the experimental set-up (I will come back to this in the next points when addressing ecological validity and the limitation of the study.)

Response. We understand the reviewer's concern. Therefore, we update the text highlighting that maintaining a minimum PTTC level is a sufficient condition for collision avoidance. We add and modify the following to better explain our results.

"We quantify how near a miss is using T_p and study how it changes with sidewalk width, agent type, and interaction format."

The change affects Page 4, Left column, Observations, Line 40 to Line 41.

"In addition, the e-scooter-to-e-scooter interaction has low T_p in all experiment widths"

The change affects Page 4, Right column, Line 36.

"Specifically, with current riding behaviour patterns, sidewalks under 2.0 m in width show low T_p and very close misses. Since maintaining a lower bound for PTTC is a sufficient condition for collision avoidance and not a necessary one, the near-misses, although causing pedestrian discomfort, may actually be safe. However, a rider behaviour modifying action, keeping PTTC above a certain level, guarantees pedestrian safety. Moreover, it improves pedestrian comfort and subjective safety in a shared space."

The change affects Page 8, Left column, Line 7 to Line 16.

Comment 1-d. On page 7, the paper reports "A central controller uses measured relative positions and velocities, monitors pedestrians' subjective safety, and intervenes to maintain a safe T_p , resulting in more pleasant interactions for pedestrians." If commercial AEB would act on T_p , the number of false positive would be unacceptable. Further, active safety systems rely on predictions. On a time-horizon of 1-2 s, autonomous interventions based on predicted T_p may not be appropriate. T_p might be a good indicator

for traffic simulations or used to give warnings when an e-scooterist consistently exhibits T_p below average; however, autonomous interventions based on T_p do not seem viable. Further, pedestrian heading is very hard to measure and can change instantly, creating some severe challenges for sensing and prediction, independently of the technology available. (BTW: passive ADAS sounds like an oxymoron and may require some reference for reader to understand what you mean.)

Response. Considering the reviewer’s valid concerns about the practicalities of an active ADAS based on PTTC, we removed the suggested technological upgrade mentioning the potential problems. We also give examples of the passive ADAS term.

“Passive systems measure PTTC in real-time, compare it to preset values, and warn the rider if driving aggressively. The alarm, for example, a sound alarm, a vibration, or a light indicator, encourages the rider to increase the PTTC and T_p . Active interventions like braking or steering away in case of low PTTC are also an option, requiring addressing challenges like sensing issues, pedestrian heading prediction, and maintaining balance.”

The change affects Page 8, Left column, Line 39 to Line 46.

Comment 2 - *The sample size of the dataset and the statistical analyses are not appropriate for the analyses and conclusions that the authors want to draw.*

A total of ten participants took part in the experiment; however, for some conditions only 6 participants provided data in pairs. In practice, in some conditions, one single subject was able to influence 1/3 of the whole dataset. Taking Figure 2(b) as an example, some of the box plots are made of 60 points (only 30 of them unique) of which a single participant may influence 20 (directly or indirectly). As a consequence, the confidence intervals are more affected by repetitions from the same subject than by the variability across subjects. So, how should we interpret these confidence intervals? What is the effect of counting data twice (setting D) on the confidence intervals? These points are important because the authors do not provide statistical analyses but propose to visually inspect the box plots to infer statistical significance based on the overlap of their notches. Because of human variability, a larger number of participants would be necessary to make sure the results are statistically sound. Performing statistical analyses and checking that the dataset responds to the requirements of the selected statistical test may also grant more trustworthy results and offer an opportunity to control for multiple comparisons. Limiting the number of conditions and increasing instead the number of participants may help provide more value to the reader.

Response. The reviewer’s main concern is the interaction setting D, used in Fig. 2(b), where

a participant, directly or indirectly, may influence one-third of the trials. Before addressing the issue, we highlight that the experiment in the paper has two major branches: objective safety, discussed in the observation section, and subjective safety. The observation section itself has five plots, each studying an independent feature. Interaction setting D, only used in Fig. 2(b), focuses on e-scooter to e-scooter interaction. The conclusions regarding pedestrian safety use the collected data from the other Interaction settings and are independent of Interaction setting D. Thus, Interaction setting D may be removed from the paper without damaging the main conclusions. However, since it may provide insights for the readers, we modify the text to alarm the reader regarding its small sample size and generalization limitation.

Regarding the Interaction setting D, we remove the duplication to avoid statistical complications, specifically arising from the independence of sampled data, and replace Fig. 2(b). The mean, median, whiskers, and quartiles do not change due to the duplication removal. The standard deviation change is so small that it did not affect Table 3. In addition, we add a note in the paper that due to the limited number of participants, this figure serves as a general guideline. We also modify the interaction setting D description in the methods section mentioning why we avoid duplication.

“Because the number of trials in Interaction setting D is limited, Fig. 2(b) serves as a general guideline.”

The change affects Page 4, Right column, Line 6 to Line 8, and Figure 2’s caption.

“Each pair performs ten trials. Although the interaction is symmetric, we did not count each trial twice to keep the data sample independent. The process results in 90 trials; $S_s=90$.”

The change affects Page 8, Interaction setting D, and Tables 2 and 3.

In addition, we used notched box plots because of their readability and compactness in the paper. At a significance level of 5%, boxes with non-overlapping notches have distinct medians. Although the level of significance relies on the assumption of a normal distribution, comparisons of medians are generally robust for other distributions. Comparing box plot medians is like a visual hypothesis test, analogous to the t-test used for means. A detailed statistical analysis as supplementary material provides the plots’ details. In the statistical analysis attachment, we check the conditions for Welch’s two-way t-test for unequal variances and unequal sample sizes and present the results for each comparable pair in the box plots in Fig. 2.

We add the following to the Fig. 2 caption.

“For a detailed comparison, see the Supplementary Tables 2-7.”

The change affects Supplementary Tables.

Comment 3 - *Some aspects of the methodology are confusing and need clarification.*

Comment 3-a. The use and calculation of PTTC may be further backed-up. Comparing PTTC to TTCA (see above) and other surrogate safety metrics may help. In addition, showing how PTTC changes over time as the road users meet may help the reader appreciate the advantages (and disadvantages) of this metric. Further, the process for extracting trajectories from stereophotogrammetry should be explained. Here, one of the critical aspects is the heading estimation; especially for pedestrians that may walk in one direction while rotating their head in another. (BTW: typically, in Nature Communications, availability of code and data are addressed.)

Response. We added Fig. 1(b) to introduce TCA. Moreover, new Fig. 1(c) and (d) show the evolution of PTTC and TCA in the hypothetical case and an experiment trial.

The change adds Fig. 1(b)-(d).

Regarding the head rotation, we used the installed markers' geometric center as the pedestrian position, and pure in-plane rotation does not change the geometric center. But, out-of-plane rotations and sideways head movements while walking change the measurements. However, since our method is line-of-sight based, we did not compensate for such movements. Because such movements affect the line of sight, PTTC should count them, especially when dealing with subjective safety.

To further explain the stereophotogrammetry, we add the following to the methods section.

“We install multiple groups of retro-reflective markers on the pedestrians' helmets, Fig. 5(a) to Fig. 5(c), and on the scooter, Fig. 5(d). **The markers in each group form a unique 3D configuration, enabling the motion capture system to locate their geometric center as the agents' relative position to a predefined origin. Therefore, the participants' head rotation does not significantly affect the position as long as the geometric center does not change. However, the head moves sideways with each step as the pedestrian walks. We did not average/compensate for the lateral movement since our method is based on pedestrian line of sight changing with the head's instantaneous position.**”

The change affects Page 9, Right column, Line 39 to Line 48.

The collected data is available in the supplementary material of the revised version.

“**All data generated or analysed during this study are included in this published article and its supplementary information files, and also are available from the corresponding author upon request.**”

“We provide the collected information in Supplementary Data.”

The change affects Page 11, Data availability and Page 10, Right column, Line 1 to Line 2.

Comment 3-b. Verification trials puzzle me a bit. I understand the concerns of the authors about verifying their metrics in a more realistic situation than the “single-encounter ones” used in the main protocol; however, I wonder what these verification trials verified. The position and behavior of the five pedestrians seems crucial in this situation and hard to control. I am not sure that these trials add much to the paper, because they raise questions that hard to answer; for instance: “How did you randomly position the five people?” and “How does this condition compare to reality?.”

Response. The presence of other pedestrians affects the results. Without the evaluation trials, we do not know if the results are generalizable to multiple pedestrian cases and what the extent of the generalizability is. We perform the evaluation trials to understand to what extent the presence of others weakens the correlations. Therefore, we compare the results between single and multiple pedestrian trials to show the effect of the presence of others on T_p and its correlation with the reported discomfort.

We add the following to the introduction to clarify the rationale.

“Moreover, we evaluate the results with an e-scooter moving through multiple pedestrians. The presence of other pedestrians shows the extent of the single encounter results’ generalizability.”

The change affects Page 2, Right column, Line 21 to Line 24.

In addition, for the strictness of the text, we change controlled and verification trials to “isolated trials and “crowd trials”. Moreover, we change randomly positioned to “arbitrarily positioned”.

The change affects multiple locations all over the paper including Fig. 2(d)-(e).

Comment 4 - *The limits of the study should be at least mentioned.*

Like all studies performed in controlled environments and limiting the experimental protocol for safety concerns, this study also has limitations that influence the ecological validity of the results. Such limitations should be mentioned and claims such as “realistic sidewalks” should be avoided. (It is hard to defend that a painted line on tarmac is a realistic sidewalk.) Although still artificial, your own “verification trials” hint already that road-user behavior will be different than in the experimental conditions as soon as other road-users will be around. Indeed, the Discussion section would benefit from discussing the results in the light of previous literature and addressing the limitations of the

study, (while limiting some of the claims about the applicability of the results that are too stretched).

Response. We add and organize the limitations of the experiments in the Methods section.

“A few factors may limit the statistical significance and generalizability of the results.”

The change affects Page 10, Right column, Line 3 to Line 4.

“Statistical significance tests assume the normality of the sample distribution. The statistical normality tests confirm the normality of the samples for most cases; see Supplementary Table 1. However, in some scenarios, the data are not normal, perhaps because of the forced top speed limit for safety and the natural minimum speed limit to maintain the balance. The deviation from normality limits the credibility of the significance tests. Moreover, presenting curbs/walls as marked lines on the experiment hallway floor affects the rider’s behaviour. We believe the riders keep more distance from curbs/walls than lines on the ground, getting closer to the pedestrians and resulting in lower T_p . In addition, the participants during the experiment are vigilant, while on actual walkways, may be inattentive or distracted and likely to behave more dangerously. Furthermore, repeated trials and switching roles can impact riding behavior through experience accumulation, too. However, since the gained experience tends to modify the riding behavior toward safer habits, it does not affect the conclusions regarding the need for behavior modification; even the safer behavior has low PTTC.”

The change affects Page 11, Left column, Line 5 to Line 30.

We also modified the language to match the experiments and the results better and limited the applicability. Specifically, we mentioned the challenges of applying the results to active ADAS.

“Active interventions like braking or steering away in case of low PTTC are also an option, requiring addressing challenges like sensing issues, pedestrian heading prediction, and maintaining balance.”

The change affects Page 8, Left column, Line 39 to Line 46.

In addition, we rephrase the mobile robot integration application as our future work.

“Furthermore, our ongoing research focuses on the imminent integration of mobile robots into public spaces, where the fitted functions are viable candidates to extend the pedestrian subjective safety estimations to mobile robots interacting with people.”

The change affects Page 8, Right column, Line 2 to Line 6.

Comment 5 - *The paper should be reorganized to help the reader understand and digest its content.*

For instance, results, methods, and discussion are mixed, making it hard for the reader to find the content she is looking for. As an example, the first part of the Results section is about methods. Further some results are interpreted and mixed with anecdotal evidence. This is an example: “In addition, during facing interactions average pedestrian velocity at the critical point is 1.5 m/s, while during the overtaking trials, it is 1.3 m/s. Interestingly, the pedestrian moves faster to avoid the e-scooter in the line of sight, while during the overtaking trials, it doesn’t have any clue of the approaching scooter and moves as usual. The difference is also partly because the e-scooter rider, accounting for the pedestrian’s possible sudden movements, approaches the pedestrian more carefully during overtaking than facing trials.”

Response. The first part of the Results section introduces the PTTC and T_p . We believe that understanding the PTTC is a prerequisite to understanding the results. Therefore, the reader can better understand the results with the current order. However, if the reviewer disagrees, we will move it to the Methods section in the next revision.

Regarding the mentioned text, we remove it from the paper, as it raises more questions and does not contribute to the paper’s novelty.

◇ Response to Reviewer 2's Comments

The authors would like to thank the reviewer for the valuable comments and suggestions for improving the paper. Next, we review the reviewer's reply point-by-point (we divided the comments into subsections for easier response tracking). Moreover, the updates in the paper are highlighted and the change's location is addressed.

Comment 0

The growth in use of e-scooters in cities around the world and the challenge of protecting both riders and pedestrians means that the current research addresses a topic which is of widespread interest and concern.

Response. We are grateful for the reviewer's understanding of the extent of the interest in sidewalk safety with the incoming new users. We hope we can address the respected reviewer's concerns.

Comment 1

The authors claim in the introduction that e-scooters are mainly used for first/last mile transport, connecting transport hubs to final destinations and they quote Ma et al. (2022). That article focused on this type of trip in one location (Washington DC) and so does not provide a strong basis for the claim. The authors neglect a range of other studies which demonstrate that many shared e-scooter trips are for fun/recreation, rather than first/last mile transport. In the same paragraph, the Clabaux et al. (2014) article is about motorcycles, not e-scooters and so it is unclear how it is relevant.

Response. Thank you for the comment. Reference [1] supports the reviewer's comment. Thus, we update the first paragraph by adding recreational uses. We also removed the irrelevant reference and the related text.

“Liu et al. categorize the growing interest into recreational or joy riding and non-recreational uses. Non-recreational applications mainly address the first/last mile of transport and connect transportation hubs to final destinations such as schools and parks; see Ma et al..”

[1]. M. Liu, J. K. Mathew, D. Horton and D. M. Bullock, "Analysis of Recreational and Last Mile E-Scooter Utilization in Different Land Use Regions," 2020 IEEE Intelligent Vehicles Symposium (IV), Las Vegas, NV, USA, 2020, pp. 1378-1385.

The change affects Page 1, Left column, Line 3 to Line 8.

Comment 2

The term “Perceived Time-To-Collision” may be confusing for some readers. When I read the title of the manuscript, I assumed that it meant a subjective measure of TTC, analogous to the use of “perceived” in “perceived safety”. The potential for confusion is exacerbated because the manuscript is comparing “PTTC” to subjective judgements of safety. I would strongly recommend changing the term. One alternative would be to keep “PTTC” but change it to stand for “Pedestrian Time-To-Collision”.

Response. Thank you for the recommendation. We understand the reviewer’s concern and changed the term to “Projected Time-To-Collision”, describing the modification in the traditional Time-To-Collision. Another option is “Pedestrian Time-To-Collision”. If the reviewer feels the latter is more appropriate, we will update it in future revisions.

“We extend TTC to 2-dimensional space and call it **Projected Time-To-Collision (PTTC)** since the extension roots in the pedestrian perception of approaching agents, and we formulate it by projecting TTC on the line of sight.”

The change affects the title, the abstract, Page 2, Right column, Line 14, and the Results section’s first paragraph.

Comment 3

Comment 3-a. Figure 2 – it would be useful to alert the reader to the discrepant scales across the five panels.

Response. We add the following note to the figure caption.

“**The scale changes across the sub-figures. For plot details, see the Supplementary Table 1.**”

The change affects the Fig. 2 caption.

Comment 3-b. Were lateral position and speed correlated?

Response. On narrow sidewalks, they are mildly correlated; the wider the sidewalk, the weaker the correlation. We add the following to page 6 to report the observation.

“**On a separate note, the lateral distance and speed are mildly correlated on narrow sidewalks, but the correlation disappears on wider sidewalks. In facing trials, the correlation R^2 s are 0.51, 0.50, 0.05, 0.00, and 0.10 for widths 1.3 m, 1.6 m, 2.0 m, 2.4 m, and 3.0 m, respectively; In the passing trial with 3.0 m width, $R^2=0.25$.**”

The change affects Page 6, Right column, Line 24 to Page 7, Left Column, Line 3.

Comment 3-c. Do you expect the relationship between PTTC and subjective safety/discomfort to be uni- or bi-directional? The subjective safety/discomfort was only for the pedestrian as far as I can gather.

Response. Although the PTTC is bilateral between the e-scooter rider and the pedestrian, the rider's reported subjective safety does not correlate with T_p . An explanation is that other parameters, for example, balancing the e-scooter or movement restrictions due to non-holonomic constraints, affect the rider's discomfort and blur the T_p role. For completeness of the report, we add the following text.

Although the PTTC is bilateral between the e-scooter rider and the pedestrian, the rider's reported subjective safety does not correlate with T_p . An explanation is that other parameters, for example, balancing the e-scooter or movement restrictions due to non-holonomic constraints, affect the rider's discomfort and blur the T_p role."

The change affects Page 6, Left column, Line 3 to Line 9.

Comment 3-d. The comments in the results section about the lack of relationship between PTTC and discomfort suggests that the authors assume PTTC underlies discomfort. Was level of discomfort higher in overtaking than facing?

Response. On a sidewalk with a 3.0 m width for the facing trials, Fig. 3(a), the average reported discomfort is 1.47 with a standard deviation of 1.24 compared to the overtaking trials, Fig. 3(b), with an average of 1.57 and a standard deviation of 1.22. A two-sample t-test fails to reject the null hypothesis with $p=0.57$ and a test power of $1-\beta=0.09$. The same trials in Fig. 2(c) demonstrate higher T_p for the overtaking case. While the high T_p corresponds to low reported discomfort in the facing scenario, the overtaking trials do not follow the pattern. The reported discomfort correlates with the T_p in facing trials. In overtaking cases, it does not correlate with T_p , suggesting other contributing factors dominate the reported discomfort. For completeness of the report, we add the following text to the paper.

"In addition, the pedestrians' reported discomfort difference between the facing and the overtaking trials is not statistically significant. On a sidewalk with a 3.0 m width for the facing trials, Fig. 3(a), the average reported discomfort is 1.47 with a standard deviation of 1.24 compared to the overtaking trials, Fig. 3(b), with an average of 1.57 and a standard deviation of 1.22. A two-sample t-test fails to reject the null hypothesis with $p=0.57$. The same trials in Fig. 2(c) demonstrate higher T_p for the overtaking case. While the high T_p corresponds to low reported discomfort in the facing scenario, the overtaking trials do not follow the pattern. The reported discomfort correlates with the T_p in facing trials. In overtaking cases, it does not correlate with T_p , suggesting other contributing factors dominate the reported discomfort."

The change affects Page 7, Right column, Line 1 to Line 14.

Comment 3-e. Comparing the values obtained in the "control" scenarios with the more

realistic multi-pedestrian scenario E may not be appropriate given that pedestrians may feel greater discomfort because they perceive they have less freedom to move away from the e-scooter (because of the presence of other pedestrians). On the other hand, the e-scooter rider may slow down more after passing the first pedestrian and so there is less discomfort in this situation.

Response. The presence of other pedestrians affects the results. Without the evaluation trials, we do not know if the results are generalizable to multiple pedestrian cases. We perform the evaluation trials to understand to what extent the presence of others weakens the correlations. Therefore, we compare the results between single and multiple pedestrian trials to show the effect of the presence of others on T_p and its correlation with the reported discomfort.

In Fig. 2 (d)-(e), we compare T_p between the single and multiple pedestrian trials ignoring the induced discomfort. In Fig. 3, we compare the correlation between T_p and the reported discomfort between the trials. Since the T_p accounts for the e-scooter speed, if the e-scooter slows down, it results in higher T_p s, and the effect is accounted for in both cases.

We add the following text to page 4 to clarify the reasoning behind performing the multiple pedestrians tests.

“The presence of other pedestrians affects the results. The evaluation trials demonstrate to what extent the presence of others affects the T_p and weakens the correlations. In other words, we compare the results between single and multiple pedestrian trials to show the effect of the presence of others on T_p and its correlation with the reported discomfort in subjective safety estimation.”

The change affects Page 4, Right column, Line 40 to Line 46.

Comment 3-f. Page 6, last paragraph – the authors conclude that the PTTCs are too low in most cases for safe interaction but they didn’t report any collisions occurring during the collection of their experimental data. Doesn’t this query this conclusion?

Response. E-scooter accidents usually happen when another contributing factor is present besides aggressive behavior. The contributing factor examples are hazardous road features, pedestrians in blind spots, or a brief distraction. We focus on the riding behavior, eliminating the other contributing factors.

We compare PTTC or T_p with the required response time and observe that in most cases, there is a time span during the e-scooter-pedestrian interaction in which the PTTC becomes smaller than the response time. If other contributing factors happen during this period, there is no time to prevent the potential accident. However, the participants are

attentive, and the experimental hall has a smooth surface. Thus, the second contributing factor to accidents is not present. Therefore, we can observe aggressive riding behavior without any collisions. Moreover, observing dangerous patterns in our near-perfect environment suggests the situation is worse on sidewalks.

We add the following text to the paper to clarify the rationale for safety concerns.

“We compare PTTC or T_p with the required response time and observe that in most cases, there is a time span during the e-scooter-pedestrian interaction in which the PTTC becomes smaller than the response time. If another contributing factor, for example, hazardous road features, pedestrians in blind spots, or a brief distraction, happens during this period, there is no time to prevent the potential accident.”

The change affects Page 4, Left column, Line 19 to Line 26.

Comment 3-g. The extension of the PTTC to mobile robots may not be appropriate as a way to reduce pedestrian discomfort because pedestrians may use eye-contact with e-scooter riders which they could not with mobile robots (although I note the research into external HMIs on automated vehicles to improve the safety of interactions with pedestrians).

Response. The pedestrian discomfort is a Multi-Input Multi-Output function. Features like the interacting agents, an e-scooter or a mobile robot, shape, size, noise, and PTTC are the inputs and result in a range of interpretations of subjective safety. Eye contact and other gesture recognitions are inputs to the e-scooter MIMO function that are not present in mobile robot MIMO function. However, HMI’s may partially compensate for the absence. PTTC plays a role in both e-scooter and mobile robot interaction with pedestrians. As a starting point, we propose a simplified Single-Input Single-Output (SISO) function to quantify the interaction discomfort. Our ongoing research focuses on building upon the e-scooter experience, providing a pleasant pedestrian walk on the sidewalks along the e-scooters and mobile robots. We removed the mobile robot part of the applications to avoid stretching the ideas.

Comment 4 - Method

Comment 4-a. In Interaction setting A and C (not clear whether this happens in B), the rider and participant swap roles. Does swapping roles influence the subjective safety because of the accumulation of experience in the experiment?

Response. It does happen in Interaction setting B, too. We add that to the paper.

“With role swapping, the number of trials is identical to setting A”

The change affects Page 8, Right column, Interaction setting B.

We add the accumulation of experience in the experiments as a limitation to the Methods section.

“Furthermore, repeated trials and switching roles can impact riding behavior through experience accumulation, too. However, since the gained experience tends to modify the riding behavior toward safer habits, it does not affect the conclusions regarding the need for behavior modification; even the safer behavior has low PTTC.”

The change affects Page 11, Left column, Line 20 to Line 25.

Comment 4-b. How do participants know the sidewalk width? If there were real curbs, would this reduce subjective safety? It may have been better to conduct the experiment with physical edges to the sidewalks.

Response. The participants were told to stay inside colored lines marked on the ground, each corresponding to a specific width. Due to width changes between trials, conducting many trials using real curbs is impractical. With real curbs, we expect higher reported discomfort levels.

“The participants were told to stay inside colored lines marked on the ground, each corresponding to a specific width.”

The change affects Page 8, Right column, Line 22 to Line 24.

In addition, we highlight the reviewer’s point in the limitations.

“Moreover, presenting curbs/walls as marked lines on the experiment hallway floor affects the rider’s behaviour. We believe the riders keep more distance from curbs/walls than lines on the ground, getting closer to the pedestrians and resulting in lower T_p .”

The change affects Page 11, Left column, Line 13 to Line 17.

Comment 4-c. Does e-scooter always overtake pedestrian on the right? What is rule in Singapore – keep to the left or right?

Response. No. The e-scooter rider and the pedestrian freely choose the side to interact with each other. The riders do not have a significant preferred side. The contributing factors to side selection are the available space beside the pedestrian and the relative heading. In addition, our participants drive on the right side of the road (RHT); Singapore is LHT.

Comment 4-d. Does less than 6 months of e-scooter experience mean that some participants had a small degree of e-scooter experience and some had none? Please clarify in the text.

Response. Yes. The participants are novice e-scooter riders. Some of them had a couple of rides before and some had none. Two participants have a few months of experiment and no participant is a regular e-scooter user. We modified the text accordingly.

“All participants are 20 to 35-year-old university students and are novice e-scooter riders. Some have a couple of rides before, and some have none. Two participants have a few months of experience, and no participant is a regular e-scooter user; age and experience may affect the e-scooter rider’s behaviour as an extraneous variable.”

The change affects Page 9, Right column, Line 11 to Line 14.

Comment 4-e. The top speed of 5 m/s is consistent with the speed limits in countries with sidewalk speed limits for e-scooters as noted by the authors.

Response. Yes. It is also the manufacturer’s suggested preset option.

REVIEWER COMMENTS

Reviewer #1 (Remarks to the Author):

The new version of the paper is definitely an improvement. While convinced about the novelty of the study and the potential of the methodologies employed, I am still hesitant about the relevance of some parts of this work as for my previous comments. The authors replied to all my previous comments; however, on some points we still have different views. Below, is a short summary of the concerns that I still have in relation to my previous comments. I trust the editor to act as a referee, where needed.

Comment 1 (Some of the main arguments and conclusions in the article are not supported by the results.) has been addressed: the new Fig 1c and 1d are a great help. I still believe that "passive ADAS" is an awkward concept, though. I invite—again—the authors to adhere to the literature. To put it bluntly, if "passive ADAS" is a thing, you should be able to find at least one reference to back up your terms. In the literature, passive systems are systems that prevent injuries, e.g., airbags. ADAS, as intended in this paper, are active systems, they may be warning or intervention systems (SAE J3063).

Comment 2 (The sample size of the dataset and the statistical analyses are not appropriate for the analyses and conclusions that the authors want to draw.) I still have the same concerns about sample size, and I understand the authors cannot gather more data. At least acknowledging this aspect in the limitations of the study seems appropriate. As for the statistics, they are not rigorous but most likely they are still correct.

Comment 3 (Some aspects of the methodology are confusing and need clarification.) The authors have addressed my concerns.

Comment 4 (The limits of the study should be at least mentioned.) The authors have addressed this comment and the new part on the study's limitations reads well. However,

- 1) Please include the sample size as a limitation in your study
- 2) About the sentence "However, since the gained experience tends to modify the riding behavior toward safer habits..." Please, give a reference or take this sentence away; I do not think this is always true; otherwise, the literature on over trust and risk seeking would not exist.

Comment 5 (The paper should be reorganized to help the reader understand and digest its content.) I still feel that mixing results with methods and discussion is unusual for a scientific paper and makes it harder for the reader to search specific information without reading the whole paper.

In addition, there are some absolute (over)statements that should be smoothed out, possibly by using conditional tenses—as we do all the time in scientific writing.

Some examples:

"Regulators'solutions are inconsistent and conflicting worldwide due to a lack of widely accepted pedestrian safety metrics." – I do not think there is agreement on this statement, this is the authors' opinion. It is hard to believe that one single metric may create global agreement and consistency on regulations.

"Thus, they join sidewalks endangering pedestrians, the primary sidewalk users, and causing undue discomfort." Again, this is speculation. I am not sure this happen inevitably at all times. Do you have a reference or is it just anecdotal evidence?

Reviewer #2 (Remarks to the Author):

The authors have addressed my comments adequately.

Response to referees-Rev. 02, NCOMMS-23-56216B, “Pedestrians’ safety using projected time-to-collision to electric scooters”

Statement of Changes and Revisions

The authors would like to thank the reviewers for their valuable comments and suggestions. Here is a list of changes that are made according to the reviewers’ suggestions and the replies to the comments.

1. Reviewer 1’s Comment No. 1 leads to the following update: Page 7, Left column, Line 13 to Line 17.
2. Reviewer 1’s Comment No. 2 leads to the following update: Page 11, Left column, Line 29 to Line 36.
3. Reviewer 1’s Comment No. 4 leads to the following update: Page Page 11, Left column, Line 29 to Line 36.
4. Reviewer 1’s Comment No. 5 leads to the following updates: Page 2, Right column, Line 35 to Line 36; Page 7, Left column, Line 42 to Line 43; the abstract; Page 1, Left column, Line 16.

We further clarified issues the reviewers raised in the response letter. Please find attached the revised manuscript and a summary of the detailed responses to the reviewers.

◇ Response to Reviewer 1's Comments

The authors would like to thank the reviewer for the valuable comments and suggestions. We believe the comments in the previous revision led to the improvement of this paper, especially comparing the suggested metric evolution with a previous metric and providing a detailed statistical analysis. In the current revision, the term misuse is addressed, the sample size is added as a limiting factor, and PTTC and TCA introductions are moved to the Methods section. Next, we review the reviewer's reply point-by-point. Moreover, the updates to the previous revision are highlighted and the change's location is addressed.

Comment 0- *The new version of the paper is definitely an improvement. While convinced about the novelty of the study and the potential of the methodologies employed, I am still hesitant about the relevance of some parts of this work as for my previous comments. The authors replied to all my previous comments; however, on some points we still have different views. Below, is a short summary of the concerns that I still have in relation to my previous comments. I trust the editor to act as a referee, where needed.*

Response. We again appreciate the reviewers' comments improving the paper. We hope the further modifications/explanations in the current revision solve the remaining concerns.

Comment 1 - *(Some of the main arguments and conclusions in the article are not supported by the results.) has been addressed: the new Fig 1c and 1d are a great help. I still believe that "passive ADAS" is an awkward concept, though. I invite—again—the authors to adhere to the literature. To put it bluntly, if "passive ADAS" is a thing, you should be able to find at least one reference to back up your terms. In the literature, passive systems are systems that prevent injuries, e.g., airbags. ADAS, as intended in this paper, are active systems, they may be warning or intervention systems (SAE J3063).*

Response. We appreciate the reviewer's input on the standardized terms in safety systems. We agree that the terms in the previous revision differ from already defined terminology. Thus, we change it to a more descriptive wording to avoid confusion. We modified the following in the paper.

“Examples of technological upgrades are warning systems and active interventions; warning systems notify the rider of low PTTC, while active interventions engage by braking or steering.

Warning systems measure PTTC in real-time,...”

The change affects Page 7, Left column, Line 13 to Line 17.

Comment 2 - *(The sample size of the dataset and the statistical analyses are not appropriate for the analyses and conclusions that the authors want to draw.) I still have the same*

concerns about sample size, and I understand the authors cannot gather more data. At least acknowledging this aspect in the limitations of the study seems appropriate. As for the statistics, they are not rigorous but most likely they are still correct.

Response. We appreciate the reviewer's feedback regarding the sample size and statistical analyses in our manuscript. We agree that a larger dataset potentially strengthens the robustness of our findings. We acknowledge the concern in the manuscript's limitations section.

“While this study provides insights into pedestrians' safety and comfort when interacting with e-scooters, the sample size may limit the generalizability of the results and not capture the full spectrum of variability in the broader population. Larger sample sizes would be beneficial for obtaining more robust and representative results, allowing for greater confidence in the generalizability of the findings to the target population.”

The change affects Page 11, Left column, Line 29 to Line 36.

Comment 3 - *(Some aspects of the methodology are confusing and need clarification.) The authors have addressed my concerns.*

Response. Thank you once again for your valuable feedback in refining our manuscript.

Comment 4 - *(The limits of the study should be at least mentioned.) The authors have addressed this comment and the new part on the study's limitations reads well. However, 1) Please include the sample size as a limitation in your study 2) About the sentence “However, since the gained experience tends to modify the riding behavior toward safer habits...” Please, give a reference or take this sentence away; I do not think this is always true; otherwise, the literature on over trust and risk seeking would not exist.*

Response. We add the sample size limitation to the limitations section.

The change affects Page Page 11, Left column, Line 29 to Line 36.

In addition, we remove the not properly supported sentence.

Comment 5 - *(The paper should be reorganized to help the reader understand and digest its content.) I still feel that mixing results with methods and discussion is unusual for a scientific paper and makes it harder for the reader to search specific information without reading the whole paper.*

In addition, there are some absolute (over)statements that should be smoothed out, possibly by using conditional tenses—as we do all the time in scientific writing. Some examples:

“Regulators’ solutions are inconsistent and conflicting worldwide due to a lack of widely accepted pedestrian safety metrics.” – I do not think there is agreement on this statement, this is the authors’ opinion. It is hard to believe that one single metric may create global agreement and consistency on regulations.

“Thus, they join sidewalks endangering pedestrians, the primary sidewalk users, and causing undue discomfort.” Again, this is speculation. I am not sure this happen inevitably at all times. Do you have a reference or is it just anecdotal evidence?

Response. We move the “Projected Time-To-Collision” and “Time of Closest Approach” introductions from the Results section to the Methods section and modify the necessary parts.

The change affects Page 2, Right column, Line 35 to Line 36, Page 7, Left column, Line 42 to Line 43. In addition, “Projected Time-To-Collision” and “Time of Closest Approach” subsections are moved from the Results section to the Methods section.

Moreover, we appreciate the reviewer pointing out the overstatements. We modify the sentences as follows.

“Regulators’ solutions are inconsistent and conflicting worldwide. Widely accepted pedestrian safety metrics may lead to converging solutions.”

The change affects the abstract.

Regarding the pedestrian discomfort caused by electric scooters, we added two references supporting the sentence.

“Thus, they join sidewalks endangering pedestrians, the primary sidewalk users, causing undue discomfort, **as reported by Che et al. and Šucha et al.**”

The change affects Page 1, Left column, Line 16.

◇ **Response to Reviewer 2's Comments**

Comment 0

- The authors have addressed my comments adequately.

Response. The authors would like to thank the reviewer for the valuable comments and suggestions for improving the paper.